# Improving Met Office seasonal predictions of Arctic sea ice using assimilation of CryoSat-2 thickness

Edward W. Blockley[1] and K. Andrew Peterson[1]

[1]Met Office, FitzRoy Road, Exeter, EX1 3PB, United Kingdom

*Correspondence to*: Ed Blockley (ed.blockley@metoffice.gov.uk)

**Abstract.** Interest in seasonal predictions of Arctic sea ice has been increasing in recent years owing, primarily, to the sharp reduction in Arctic sea ice cover observed over the last few decades, which is projected to continue. The prospect of increased human industrial activity in the region, as well as scientific interest in the predictability of sea ice, provides important motivation for understanding, and improving, the skill of Arctic predictions. Several operational forecasting centres now routinely produce seasonal predictions of sea ice cover using coupled atmosphere-ocean-sea ice models. Although assimilation of sea ice concentration into these systems is commonplace, sea ice thickness observations, being much less mature, are typically not assimilated. However many studies suggest that initialisation of winter sea ice thickness could lead to improved prediction of Arctic summer sea ice. Here, for the first time, we directly assess the impact of winter sea ice thickness initialisation on the skill of summer seasonal predictions by assimilating CryoSat-2 thickness data into the Met Office's coupled seasonal prediction system (GloSea). We show a significant improvement in predictive skill of Arctic sea ice extent and ice-edge location for forecasts of September Arctic sea ice made from the beginning of the melt season. The improvements in sea ice cover lead to further improvement of near-surface air temperature and pressure fields across the region. A clear relationship between modelled winter thickness biases and summer extent errors is identified which supports the theory that Arctic winter thickness provides some predictive capability for summer ice extent, and further highlights the importance that modelled winter thickness biases can have on the evolution of forecast errors through the melt season.

## 1 Introduction and motivation

Arctic sea ice is one of the most rapidly, and visibly, changing components of the global climate system. The past few decades have seen a considerable reduction in the extent and thickness of Arctic sea ice (Vaughan et al., 2013; Meier et al., 2014; Lindsay and Schweiger, 2015; Kwok et al., 2009). Although the areal extent of Arctic sea ice has declined in all seasons, the reduction has been most pronounced in the summer with the seasonal minimum extent hitting record low values in September 2007 and 2012 (Meier et al., 2014; Vaughan et al., 2013). This decline is projected to continue in the future in response to rising global temperatures and atmospheric $CO_2$ concentrations (Collins et al., 2013; Notz and Stroeve, 2016).

In response to declining sea ice cover, human activity in the Arctic is increasing with access to the Arctic Ocean becoming more important for socioeconomic reasons (Meier et al., 2014). Such activities include commercial activities like tourism, fishing, mineral and oil extraction, and shipping (Smith and Stephenson, 2013), along with activities of importance to local communities such as subsistence hunting and fishing, search and rescue, and community re-supply. Accurate forecasts of Arctic sea ice are therefore becoming increasingly important for the safety of human activities in the Arctic (Eicken, 2013). Improved knowledge of sea ice on seasonal timescales allows for better planning which should lead to a reduced level of risk and a reduction in operational costs for human activities in the Arctic Ocean. Regional changes in Arctic sea ice cover can also have implications for lower-latitude weather and climate (Koenigk et al. 2016; Balmaseda et al. 2010; Screen, 2013). For example, Koenigk et al. (2016) show that late summer sea ice cover can be linked to winter North Atlantic Oscillation (NAO)-like patterns and blocking in Western Europe. More accurate Arctic sea ice predictions therefore, can also contribute to improved forecasts, and hence longer-term planning, in mid-latitude regions.

Interest in seasonal predictions has increased following the drastic reduction in Arctic sea ice extent in the summer of 2007, which led to a (then) record-low summer minimum extent being set. In response to this, in 2008, the Sea Ice Outlook (SIO) was instigated by the Study of Environmental ARctic CHange (SEARCH) to synthesise seasonal predictions of September Arctic sea ice extent, made from late spring and early summer, using a variety of modelling, statistical, and heuristic approaches (see Stroeve et al., 2014). For seasonal forecasts to be of use to stakeholders, a thorough understanding of their predictive skill is needed. The community that has been built up around the SIO has enabled collaborative activities addressing such issues across various prediction centres through the inter-comparison and common evaluation of forecasts (see https://www.arcus.org/sipn/sea-ice-outlook). There is also an interesting scientific problem here to test our ability to predict sea ice on seasonal timescales that are considerably longer than the (typically 1-2 week) limit, beyond which, the chaotic nature of the atmosphere and ocean inhibit traditional deterministic forecasting (Slingo and Palmer, 2011). As the sea ice thins, variability in ice extent increases (Holland et al., 2011; Goosse et al., 2009) and so the problem of making seasonal Arctic sea ice predictions – particularly for the September minimum – is one that is getting more challenging and interesting as the ice cover declines (Holland et al., 2011; Stroeve et al., 2014).

Although global coupled forecasting systems have been used successfully for seasonal prediction of mid-latitude weather and climate for some time now (see for example Scaife et al., 2014), their application to Arctic sea ice prediction is much less mature. In particular, forecasts in the Arctic are hampered by the fact that observations are much less abundant and data assimilation techniques less advanced in the polar regions than at lower-latitudes (Jung et al., 2016; Bauer et al., 2016), meaning that the initial conditions used for forecasts in the Arctic are less accurate than for lower latitudes. Despite this, several operational forecasting centres regularly contribute to the SIO with sea ice predictions from fully coupled atmosphere-sea ice-ocean modelling systems. One such system is the Met Office's Global Seasonal (GloSea) coupled ensemble prediction system (MacLachlan et al., 2014; Peterson et al., 2015) which has contributed to the SIO since 2010. The ocean and sea ice components

of GloSea are initialised each day using the Forecast Ocean Assimilation Model (FOAM) operational ocean-sea ice analysis of Blockley et al. (2014; 2015). FOAM routinely assimilates sea ice concentration along with various ocean quantities (satellite and in-situ Sea Surface Temperature (SST), satellite Sea Level Anomaly (SLA), in-situ profiles of temperature and salinity) but, in common with most operational ocean analysis systems (Tonani et al., 2015; Martin et al., 2015; Balmaseda et al., 2015; Cummings and Smedstad, 2014), does not assimilate sea ice thickness.

The use of dynamical models for seasonal sea ice prediction is in its relative infancy. Still, there have been several studies that demonstrate skill in retrospective forecasts (or hindcasts) of September-mean Arctic sea ice extent made from spring (e.g. Sigmond et al., 2013; Wang et al., 2013; Chevallier et al., 2013; Msadek et al., 2014; Peterson et al., 2015). However, none of these were able to match the potential skill found in idealised "perfect model" studies (Guemas et al., 2016; Tietsche et al., 2014; Day et al., 2014; Blanchard-Wrigglesworth et al., 2011), where all the initial conditions, but in particular, the sea ice thickness, are known precisely. Furthermore, when applied to a real-time forecast, as submitted to SIO, the skill was found to be even lower than the hindcast skill (Blanchard-Wrigglesworth et al., 2015), and only marginally better than a linear trend forecast (Stroeve et al., 2014). Clearly, there is potential for improvement in the dynamical models if more complete initial conditions are known – with an even greater need, as demonstrated by the deteriorated performance of the real-time forecasts, for more accurate real-time initial conditions. None of the systems mentioned above initialise the sea ice using observed thickness measurements.

Several studies have shown that winter sea ice thickness provides important preconditioning for the evolution of Arctic sea ice through the summer melt season. Blanchard-Wrigglesworth and Bitz (2014) found sea ice thickness anomalies in general circulation models (GCMs) to have a timescale of between 6 and 20 months making their correct representation in model initial conditions of importance for seasonal predictions. Other modelling studies by Holland et al. (2011) and Kauker et al. (2009) have also shown that knowledge of winter ice thickness can provide some predictive capability for summer ice extent. Perfect model studies (e.g. Day et al., 2014) have also suggested that correct initialisation of sea ice thickness can lead to improved seasonal forecasts. Day et al. (2014) used the HadGEM1.2 climate model to show that memory of winter thickness conditions can persist well beyond seasonal timescales and provide predictive capability for up to 2 years. Collow et al. (2015) found considerable changes in ice concentration forecasts when changing the initial thickness in the coupled forecast system model version 2 (CFSv2) seasonal prediction system. They showed an improvement to seasonal forecasts when using thickness fields from the Pan-Arctic Ice-Ocean Model Assimilation System (PIOMAS) model of Zhang and Rothrock (2003). These studies suggest that seasonal (> 90 days) predictions of Arctic summer sea ice, made with dynamical models, could be improved by correctly initialising sea ice thickness. However, we note that, although uncertainty in the initial conditions plays a crucial role for seasonal predictions of Arctic sea ice, model uncertainty is likely to dominate the evolution of seasonal forecast errors (Blanchard-Wrigglesworth et al., 2017).

Although the assimilation of sea ice concentration has been included in ocean reanalysis, operational ocean prediction and seasonal forecasting systems for several years (Stark et al., 2008; Peterson et al. 2015), sea ice thickness is not yet routinely used to initialise these systems (Martin et al., 2015; Balmaseda et al., 2015; Tonani et al., 2015). There have however, been several recent studies that have sought to improve the representation of Arctic sea ice thickness in analyses and short-range

forecasts using satellite thickness products derived from Soil Moisture and Ocean Salinity (SMOS) brightness temperatures and/or from CryoSat-2 (hereafter CS2) radar freeboard measurements. Such studies have generally focused on assimilation of thickness using ensemble techniques into short-range, externally forced, ocean-sea ice models in the Topaz system (Xie et al., 2016) or using MITgcm (Yang at al., 2014; Mu et al., 2018). Although these studies showed considerable improvement to the simulation of sea ice thickness, the impact on short-range forecasts of sea ice concentration or extent was minimal. More

recently, Allard et al. (2018) used direct initialisation of CS2-derived sea ice thickness, using 2 different datasets processed with different algorithms, within a series of reanalyses performed with the Naval Research Laboratory's forced ocean-sea ice Arctic Cap Nowcast/Forecast System (ACNFS). They show that the analysed sea ice thickness is significantly improved when assimilating CS2 thickness compared against in-situ and airborne measurements. They also perform an in-depth assessment of the thickness data and analyses, and show a good agreement between the CS2-derived ice thickness and observations from in-

situ and airborne sources.

As noted above, several studies (Yang at al., 2014; Xie et al., 2016; Mu et al., 2018; Allard et al., 2018) have looked at the impact of sea ice thickness initialisation on analyses and short-range forecasts produced with externally forced ocean-sea ice models. What has not been investigated is the impact that assimilation of sea ice thickness may have in longer (> 90 days),

forecasts made using fully coupled dynamical models. Here we do so for the first time using the Met Office GloSea coupled seasonal prediction system. For accurate seasonal predictions of September sea ice cover, it is important to model ice that will persist throughout the summer season, and so an improved representation of the location of thick sea ice within the initialisation should be advantageous. We hypothesise that GloSea seasonal predictions of late-summer (September) sea ice cover will be improved by initialising sea ice thickness in early spring (May), using observations of thick sea ice derived from CS2. In this

study, we use a simple nudging technique to test this hypothesis, and evaluate the feasibility of including sea ice thickness initialisation within the operational GloSea seasonal prediction system. We assimilate CS2 sea ice thickness within the FOAM ocean-sea ice reanalysis and use these analyses as initial conditions for an ensemble of seasonal (5-month) coupled forecasts to determine the impact of sea ice thickness initialisation on the skill of GloSea seasonal predictions. We show that sea ice thickness initialisation leads to a considerable improvement in the skill of seasonal predictions of Arctic sea ice extent and ice

edge location.

This paper is structured as follows: Section 2 introduces the modelling systems and observations used in this study; Section 3 describes the initialisation of CS2 thickness within the ocean-sea ice reanalysis and the generation of initial conditions for seasonal predictions; Section 4 provides details of GloSea coupled seasonal prediction experiments performed using CS2

**Deleted:** NRL

**Deleted:** In this study, we use a nudging technique to assimilate CS2 sea ice thickness within the FOAM/GloSea reanalysis system and use these initial conditions to determine the impact of sea ice thickness initialisation on the skill of GloSea seasonal predictions. Although

**Deleted:** term

**Deleted:** and in

**Deleted:** , there have been no studies exploring the impact of initialising coupled seasonal forecasts using sea ice thickness.

**Deleted:** and

**Deleted:** assimilation

**Deleted:** and the impact on

**Deleted:** forecast

initialised thickness and shows improved skill for seasonal forecasts of Arctic ice cover. Section 5 provides summary discussion and an overview of proposed future work.

## 2 Models and observations used in this study

### 2.1 Modelling systems

The model systems used in this study are taken from the Met Office suite of seamless, traceable prediction systems introduced by Brown et al. (2012) using components of the Hadley Centre Global Environment Model version 3 (HadGEM3) coupled model architecture described by Hewitt et al. (2011). All of these HadGEM3-based modelling systems simulate the ocean and sea ice conditions using the Nucleus for European Modelling of the Ocean (NEMO) ocean model (Madec, 2008) coupled to the Los Alamos sea ice model (CICE) (Hunke et al., 2015).

Within the Met Office's unified, seamless framework, seasonal forecasts are performed using the GloSea coupled prediction system (MacLachlan et al., 2014; Scaife et al., 2014). GloSea produces two 210-day seasonal forecasts every day, which, together with those from previous days, are combined to form a lagged ensemble prediction system. Meanwhile hindcasts – retrospective forecasts performed for previous years using true forecast conditions – are used to establish errors in the model
climatology for the purposes of bias correction, and to estimate forecast skill. More details on the GloSea seasonal prediction system can be found in MacLachlan et al., (2014). The ocean and sea ice components of the GloSea system are initialised each day using analyses from the FOAM system described in Blockley et al. (2014; 2015). FOAM is an operational ocean-sea ice analysis and forecast system run daily at the Met Office. Satellite and in-situ observations of temperature, salinity, sea level anomaly and sea ice concentration are assimilated by FOAM each day using the NEMOVAR 3D-Var First Guess at
Appropriate Time (FGAT) scheme. Sea ice thickness is not currently assimilated in FOAM; new ice is added by the concentration assimilation at a default thickness of 0.5 metres. More details of the FOAM system can be found in Blockley et al. (2014) and more about the NEMOVAR 3D-Var FGAT scheme used therein can be found in Waters et al. (2015).

As well as the abovementioned operational analyses and forecasts, longer reanalyses are performed with the FOAM system
using surface forcing derived from the ERA-Interim atmospheric reanalysis (Dee et al., 2011). Within the GloSea seasonal prediction system, hindcast experiments initialised from these reanalyses are used to bias correct the GloSea seasonal forecasts (see MacLachlan et al., (2014) for more information). As well as being used for bias correction within GloSea, these ocean reanalyses are utilised more widely within the ocean community (Balmaseda et al., 2015; Chevallier et al., 2017; Uotilla et al., 2018) and have been used to help answer a number of other scientific questions (e.g. by Roberts et al., 2013; Jackson et al.,
30  2015).

**Deleted:** Forecast Ocean Assimilation Model (

**Deleted:** )

**Deleted:** used

Throughout this study we shall use prototype FOAM and GloSea systems based on the latest configuration of the Met Office coupled modelling system (GC3: Williams et al., 2017) which will be used as part of Met Office Hadley Centre's contribution to phase 6 of the Coupled Model Intercomparison Project (CMIP6). This GC3 coupled model version uses the GO6 ocean and GSI8 sea ice component configurations described in Storkey et al., (2018) and Ridley et al., (2018) respectively and uses the extended ORCA025 tripolar grid described therein – with nominal 1/4° horizontal resolution, ranging from 8.9 km to 15.5 km in the Arctic Ocean basin, and 75 vertical levels. The sea ice component of the model is based upon CICE vn5.1.2 and uses the standard CICE elastic–viscous–plastic (EVP) rheology for modelling the sea ice dynamics (Hunke et al., 2015). Growth and melt of the sea ice is calculated using a multi-layer thermodynamics scheme with 4 layers of ice and 1 layer of snow. At each model grid point, the sub-grid scale ice thickness distribution is modelled by partitioning the ice into five thickness categories (lower bounds: 0, 0.6, 1.4, 2.4 and 3.6 m), with an additional ice-free category for open water areas. The impact of surface meltwater on the sea ice albedo is explicitly represented by the prognostic evolution of melt ponds using the topographic formulation. Further details about the sea ice component, and the wider coupled model used here, can be found in Ridley et al., (2018) and Williams et al., (2017) respectively.

## 2.2 Observations of sea ice thickness

Whilst observations of sea ice concentration providing large-scale coverage for both poles have been available since 1979 (Fetterer et al., 2016; Rayner et al., 2003), measurements of sea ice thickness are, relatively, much less abundant. However, satellite estimates of winter thickness have been available for a number of years using radar altimetry (Laxon et al., 2003), laser altimetry (Kwok and Cunningham, 2008), and, more recently for thin ice, microwave brightness temperatures (Kaleschke et al., 2016). Although radar altimeter estimates of sea ice thickness have been available for many years now, their up-take into operational ocean-sea ice assimilation systems has been minimal. The primary reasons for this are three-fold: the data were not made available in near-real-time for use in operational analysis systems; owing to the orbit inclination, these datasets often have a large 'pole-hole' giving poor coverage in the central Arctic Ocean; there is considerable uncertainty associated with these estimates of ice thickness (Ricker et al., 2014). The problems outlined above have been ameliorated somewhat during the last few years by the launch of ESA's CryoSat-2 satellite (CS2) whose primary objective is to acquire accurate measurements of sea ice thickness. CS2 has an unusually high inclination orbit that provides observational coverage up to 88°N, which has considerably reduced the size of the pole-hole (Laxon et al., 2013). CS2 is also fitted with a Synthetic Aperture Interferometric Radar Altimeter (SIRAL) instrument that has a higher accuracy, and along-track resolution, than was previously available from the ENVISAT and ERS-1/2 radar altimeters (Guerreiro et al., 2017). The processed data from CS2 is also provided in almost near-real-time by the Centre for Polar Observation and Modelling (CPOM) (Tilling et al., 2016) making its use within operational analysis systems a realistic proposition.

### 2.2.1 CryoSat-2 thickness observations

In this study, we initialise the model using thick ice from CS2, which are accurate for ice thicker than 1m (Ricker et al., 2017). We use monthly CS2 winter (Oct-Apr) thickness estimates produced by CPOM (Tilling et al., 2016) which start from October 2010 until present (at time of writing). Sea ice freeboard is inferred from radar altimetry aboard the CS2 satellite and is converted to thickness by assuming that the sea ice floats in hydrostatic equilibrium and by making various assumptions about the snow loading and the relative densities of the sea ice, the ocean and the overlying snow. Details of the methods used to generate the CPOM thickness fields can be found in Laxon et al., (2013) and Tilling et al. (2015). Several different centres, including CPOM, are now producing CS2-derived estimates of sea ice thickness. More details on the differences between these observational estimates can be found in Stroeve et al. (2018) and Allard et al. (2018). Some more general discussion of the uncertainties involved in the calculation of sea ice freeboard and thickness using radar altimetry can be found in Ricker et al. (2014).

The CPOM thickness data are provided on a 5 km polar stereographic grid having been smoothed with an averaging window of radius 25 km. We apply a further quality control (QC) to the data before use. The CS2 thickness retrieval methodology is particularly uncertain for thin ice where the ice freeboard is not much higher than sea level (Ricker et al. 2014; 2017). To avoid high observational error associated with these thin measurements we impose a minimum thickness threshold of 1 m – a choice that was motivated by Figure 2b of Ricker et al. (2017). Further, to ensure that the observations are as representative of the month as possible we apply the constraint that at least 10 different altimeter tracks are used to determine the monthly-mean observation. We also impose a constraint on the spread of the track observations by keeping monthly observations only when the standard deviation of the contributing individual track observations is less than 2 m. Finally, we remove any spuriously high observations by imposing a maximum thickness threshold of 7 m. In total, application of the abovementioned QC rejected roughly 21.5% of the original observations; about 9.4% of the observations were removed by the 1m cut-off and just over 12% were rejected by the remaining constraints. An example of the thickness observations used in this study can be seen in Figure 1a, which shows a map of average October-April Arctic thickness for 2011-2015 inferred from CS2 estimates after application of the QC process described above.

### 2.3 Observations of sea ice extent and concentration

Within this study, observations of sea ice concentration and extent from several sources are used both for evaluating seasonal predictions of Arctic sea ice, and for assimilation within the reanalyses used for initialisation of these seasonal predictions.

### 2.3.1 Sea ice concentration and extent datasets used for evaluation

Uncertainty associated with sea ice concentration and extent estimates from satellites is high (Ivanova et al., 2015) and the commonly used sea ice extent metric is nonlinear and dependent on resolution (Notz, 2014). To account for this uncertainty

---

**Deleted:** In this study, we

**Deleted:** To prevent any smearing of thickness values near the ice edge during the filtering step we impose a maximum displacement of 15 km between the average location of the raw track observations and the centre of the final grid point (Andy Ridout, pers. comm., 2017). We further remove any spuriously high observations by imposing a maximum thickness threshold of 7 m.

**Deleted:** sensitive

**Deleted:** was

**Deleted:** December, January and February

**Deleted:** ¶
Although SMOS thickness data is also available for much of the study period used here, we choose to use only CS2 data for this study. The motivation for this is that SMOS only provides information about thin ice up to about 0.5 m whilst the CS2 data is good for thicker ice above 1 m (Ricker et al., 2017). Here we are concerned with seasonal predictions of Arctic summer sea ice - for which ice less than 2 m thick tends to melt away completely (Keen et al., 2013). We therefore expect the assimilation of thicker ice to be more important for our needs and so use only the CS2 thickness observations for this feasibility study.¶

we include observational estimates from three different sources: extents calculated from the 1° gridded Hadley Centre sea Ice and Sea Surface Temperature (HadISST1.2) dataset of Rayner et al. (2003); the National Snow and Ice Data Center (NSIDC) sea ice index of Fetterer et al. (2016); and gridded sea ice concentration fields from the most recent FOAM-GloSea ocean-sea ice reanalysis. This reanalysis, performed using version 13 of the FOAM system (Blockley et al., 2015), is used within the

Copernicus Marine Environment Monitoring Service (CMEMS; http://marine.copernicus.eu/) global ocean reanalyses ensemble product (ID: GLOBAL-REANALYSIS-PHY-001-026; described in http://cmems-resources.cls.fr/documents/QUID/CMEMS-GLO-QUID-001-02). Using this CMEMS reanalysis has the benefit that it is performed on the same ORCA025 grid as the ocean-sea ice components of the GloSea seasonal forecasting system, which makes spatial comparisons easier. This reanalysis has also been evaluated thoroughly through the Ocean Reanalyses Inter-

comparison Project (ORA-IP) (see Balmaseda et al., 2015; Chevallier et al., 2017; Uotilla et al., 2018). To avoid confusion with the FOAM reanalyses performed as part of this study, and described later, we refer to this product as "CMEMS" hereafter.

**2.3.2 Sea ice concentration datasets used for assimilation**

The CMEMS reanalysis, and the reanalyses performed in this study, were performed using Special Sensor Microwave Imager/Sounder (SSMI/S) sea ice concentration data provided by the European Organisation for the Exploitation of

Meteorological Satellites (EUMETSAT) Ocean and Sea Ice Satellite Application Facility (OSI-SAF). Sea ice concentration is assimilated along with ocean data sources using the NEMOVAR 3D-Var scheme (see Blockley et al., 2014; Waters et al., 2015). Prior to October 2009, OSI-SAF's Global Sea Ice Concentration Climate Data Records (OSI-409, version 1.1) product was assimilated. When the reanalysis was run, in 2014, these data were only available up to the end of 2009 and so the OSI-SAF near-real-time (NRT) product OSI-401a was used from 25th October 2009 onwards. These two datasets have differences

in the processing of low concentration ice and near coastlines (see Section 4.2 of OSI-SAF, 2017). However, this does not cause us any concern here because our study is focussed on the CS2 era from October 2010 onwards.

**3 Initialisation of thickness in the ocean-sea ice reanalysis system**

Here we use the latest development version of the FOAM-GloSea reanalysis system that has been undertaken as part of the upgrade of GloSea and FOAM to use the latest GC3 version of the Met Office coupled model architecture (Williams et al.,

2017). Specifically here the ocean reanalysis system is using the GO6 ocean configuration described in Storkey et al. (2018) and the GSI8 sea ice configuration described in Ridley et al. (2018). We take the latest GO6+GSI8 reanalysis as our control (hereafter CTRL-RA) and modify it to include initialisation of sea ice thickness using CS2 observations (hereafter ThkDA-RA). The CTRL-RA reanalysis was run from 1992 to 2015 but here we only re-run the last 5 years – from October 2010 to the end of 2015 – to tie in with availability of CS2 thickness estimates.

Within the ThkDA-RA reanalysis, CS2 thickness data are assimilated using a basic nudging technique in which thickness fields are nudged towards the monthly gridded CS2 observations in a fashion akin to that employed by climatological relaxation schemes. All other data used within the control run (i.e., SST, SLA, T/S profiles, and SSMI/S concentration) are assimilated here too in the same manner as in the standard FOAM system (Blockley et al., 2014; 2015). The sea ice concentration observations assimilated are the same as used for the CMEMS reanalysis described in Section 2.3.2 above (i.e., OSI-401a). An overview of reanalysis experiments used in this study can be found in Table 1.

We use the monthly CPOM measurements introduced in Section 2.2 and map them onto the model grid using a standard binning technique. A linear interpolation is performed each day to get daily thickness observations from the nearest two months. Assimilation increments are created by taking a simple difference between these daily CS2 thickness observations and the daily-mean model thickness. Where no observations are present, the increments are set to zero to ensure no thickness nudging is performed. We do things this way to avoid problems arising with the sparse data and so we can keep nudging model towards CS2 thickness.

The increments are applied within the CICE model code in a similar fashion to the sea ice concentration assimilation described in Peterson et al. (2015) and Blockley et al. (2014). Thickness changes are made at each time step using the Incremental Analysis Update (IAU) method. A 5-day relaxation timescale is used and increments are only applied where the grid-cell ice concentration is above 40%. The CICE sea ice model uses multiple thickness categories to represent the sub-grid thickness distribution. To apply the thickness increments into the multi-category model we chose to nudge the grid-box-mean thickness towards observations by making changes across each of the 5 sub-grid categories - so long as there is ice present there with concentration above 1% - maintaining the initial distribution of volume between the categories. We note here that this approach is similar to that employed by Allard et al. (2018) who multiply the ice volume in each category by the grid-box-mean model-observation thickness difference. However, whilst they use direct initialisation, we use the IAU approach to incorporate changes into the model in a gradual manner and limit the potential for sudden shock in the system (Bloom et al., 1996).

**3.1 Impact of CryoSat-2 initialisation on reanalysis thickness**

Figure 1 illustrates the general impact of including CS2 assimilation within the ThkDA-RA reanalysis by showing the mean Arctic sea ice thickness, for the months when CS2 data is available (October-April), over the whole ThkDA-RA reanalysis (2010-2015). The difference plot in Figure 1d shows that the inclusion of CS2 nudging generally acts to increase the thickness of the Arctic sea ice – in particular in the Atlantic Sector north of Fram Strait, and to the north of Greenland. Comparison with the observations in Figure 1a shows that the thickness in ThkDA-RA is much more closely aligned with the CS2 data than is the case for CTRL-RA.

A comparison of sea ice volume for the CTRL-RA and ThkDA-RA reanalyses in Figure 2 confirms that the net effect of CS2 thickness nudging is an increase in sea ice thickness. We note that an increase in volume here directly implies an increase in average ice thickness because, as sea ice concentration is tightly constrained by the assimilation of sea ice concentration and sea surface temperature, the ice area between the two reanalysis simulations is virtually identical (not shown). Figure 2 shows that winter volume is increased the most by the assimilation of CS2 thickness. This is perhaps not surprising given that winter is the time when the data is available. However, there is some evidence that these winter changes also affect the summer volume, which is most pronounced in 2014 and, to a lesser extent, 2013 and 2015. In all years the volume time series shows a clear kink on 1st October when the CS2 data comes back online and begins to be assimilated in the reanalysis - although this is much less pronounced in 2014 when the summer thickness was also increased. In Figure 2, sea ice volume for the CTRL-RA and ThkDA-RA reanalyses are also compared with volume estimates from the PIOMAS model of Zhang and Rothrock (2003). The PIOMAS volume is included here purely as a reference because it is well understood and widely used for this purpose. The volume in the CTRL-RA run is much closer to PIOMAS than the ThkDA-RA run. This is expected as PIOMAS has been shown to underestimate thickness/volume in the winter compared to the CPOM CS2-derived thickness (Tilling et al., 2015; Laxon et al., 2013) – although it has been shown to compare better with laser altimeter estimates such as ICESat (Schweiger et al., 2011).

Figure 3a shows the impact of the CS2 thickness initialisation on the reanalysis end-of-winter thickness fields – that will be used in this study to initialise GloSea seasonal predictions – with the 5-year mean differences for 1st May at the end of winter when CS2 observations cease. At the end of winter it is apparent that inclusion of CS2 thickness nudging has increased sea ice thickness across much of the Atlantic sector of the Arctic (Barents, Kara and Greenland Seas). Conversely, ice thickness has been decreased in the Canadian Arctic Archipelago (CAA) and, to a lesser degree, across much of the Pacific sector (Beaufort, Chukchi and East Siberian Seas). Thickness is also increased in many of the marginal seas outside of the central Arctic such as the Bering Sea and Hudson Bay. One notable exception is the Labrador Sea/Baffin Bay where the differences form an east-west dipole with ice thickness being reduced along the Canadian coast but increased on the Greenland side.

Figure 3b shows the 5-year-mean difference in the reanalyses thickness fields at the end of summer (30th September) after 5 months of running without thickness assimilation. The impact of the CS2 nudging is an increase in sea ice thickness throughout much of the Arctic save for small patches in the East Siberian Sea and within the CAA. The pattern is broadly consistent with the differences seen at the end of winter in Figure 3a. Even after 5 months of running without the CS2 thickness nudging – although whilst still assimilating ice concentration and other ocean quantities – we can see the impact of initialising thickness through the winter. This is good news for the feasibility study because it tells us that the thickness changes are being retained by the model and not being rejected or washed out by the assimilation of other quantities such as ice concentration.

The general picture shown by the 5-year mean in Figure 3a is typical of the end of winter thickness differences seen for each of the 5 years 2011-2015 (not shown). Mean sea ice thickness across the Arctic Ocean basin has been increased by around 14% (from 2.00m to 2.27m). This increase is most pronounced in the Atlantic Sector of the Arctic (30°W-140°E) where thickness increased by around 33% (from 1.44m to 1.91m). Although mean thickness in the combined Beaufort and Chukchi Seas has decreased by 7% (from 2.32m to 2.15m), the net effect over the whole Pacific Sector of the Arctic (140°E-20°W) is an increase of 6.6% (from 2.33m to 2.49m). However, the situation is not so clear-cut for the summer case (Figure 3b) where thickness increases are much more pronounced in 2014 and 2013 (see Figure 2).

The impact of the thickness changes on the large-scale sea ice motion is negligible with monthly-mean velocities in the two experiments being virtually identical throughout the 2011-2015 period (not shown). This is consistent with the findings of Allard et al. (2018) who show little impact on ice drift in their reanalysis comparisons.

In summary, we have shown that nudging Arctic sea ice thickness to CS2 observations within the ThkDA-RA reanalysis has the net effect of increasing sea ice volume. The differences between the two reanalyses reveal a persistent bias in the thickness distribution in the model when compared with CS2 whereby sea ice is too thick on the Pacific side and not thick enough on the Atlantic side of the Arctic. There is evidence to suggest that the winter Arctic sea ice thickness/volume is an important precondition for evolution of ice through the melt season (in agreement with the current literature) because the effects of winter thickness changes imposed by the nudging are still evident at the end of the summer. Another important result to note here is that the assimilation of thickness worked well and the increments were successfully retained by the model, which bodes well for inclusion of sea ice thickness within the NEMOVAR system in the future.

**4 Initialisation of thickness in the GloSea coupled seasonal prediction system**

Seasonal forecasts of sea ice extent are made operationally by the GloSea system each day. Hindcast predictions, performed from a discrete predefined set of start dates each year, are also run within the operational suite each day and used as part of the bias correction process. These hindcast predictions are initialised using the long GloSea ocean-sea ice reanalysis (as described in Section 3) which is coupled to atmosphere initial conditions interpolated from the ERA-I reanalysis (Dee et al., 2011). In addition to being used operationally for bias correcting forecasts, seasonal hindcasts such as this are performed for testing of model configuration upgrades prior to implementation within the GloSea operational suite. As these hindcasts are used to test the expected skill of a real forecast, they are performed in a fashion that does not use any subsequent observational data after initialisation, so as not to invalidate that expectation. A recent trial of the new GC3 coupled model configuration of Williams et al. (2017) has been performed using the GloSea seasonal prediction system, which we shall use as our control (denoted CTRL-HC). The ocean and sea ice for these hindcasts are initialised using the control reanalysis (CTRL-RA) described in Section 3 and the atmosphere is initialised from the ERA-I reanalysis. As the GC3 developments include the implementation

of a new multi-layer model for terrestrial snow (see Walters et al., 2017; Williams et al., 2017) the snow fields were initialised separately from the atmosphere using a standalone version of the GC3 land surface component (Joint UK Land Environment Simulator; JULES) with ERA-I snow precipitation and data assimilation.

Here we wish to test the impact of initialising with CS2 sea ice thickness on the seasonal predictions of September sea ice extent. For this purpose, an ensemble of seasonal prediction experiments was configured that was identical to the CTRL-HC experiment except that the ocean and sea ice components were initialised from the ThkDA-RA reanalysis instead of CTRL-RA. Seasonal predictions were performed from 3 different spring start dates (25th April, 1st May and 9th May). For each of these start dates, an ensemble of 8 seasonal predictions was initialised from the same analysis fields with spread between the
members achieved by using stochastic physics (see MacLachlan et al., (2014) for more details). This methodology is identical to that used for CRTL-HC and, through a mixture of lagged and perturbed methods, provides an ensemble of 24 forecasts of September sea ice each year. These predictions were performed for 2011-2015 – each year that spring analyses are available from the ThkDA-RA ocean reanalysis. We denote this system of predictions as ThkDA-HC. Details of the GloSea coupled prediction experiments used in this study can be found in Table 2.

**4.1 Improvements to seasonal prediction of Arctic extent and ice edge location**

Results from the ThkDA-HC experiment show that the CS2 thickness initialisation has considerably improved the skill of GloSea seasonal predictions of Arctic sea ice cover. Figure 4 shows September-mean Arctic sea ice extent (upper panel) from the GloSea control ensemble (CTRL-HC; blue) and the ensemble run with initialised thickness (ThkDA-HC; pink). Predictions from each of the 24 ensemble members, initialised from the 3 April/May start-dates, are depicted by the crosses; the ensemble
mean is plotted with bold symbols and inter-connecting lines. Although the ThkDA-HC predictions only start from 2011 we plot the CTRL-HC throughout the whole period of the run from 1992-2015 to help put the, relatively short, 5-year time series into context. To assess the accuracy of the GloSea seasonal predictions, observational and reanalysis estimates of Arctic extent, from the CMEMS reanalysis, and the HadISST and NSIDC datasets (see Section 2.3), are plotted alongside the model predictions (black/grey). We note here that the difference in extent prior to 2010 between the CMEMS reanalysis and the
HadISST and NSIDC data sources apparent in Figure 4a is caused by the switch in OSI-SAF data products in October 2009, from OSI-409 version 1.1 to OSI-401a, described in Section 2.3 above. Being prior to the launch of CS2, this change does not have any impact on the results of our study but we include all years available from CTRL-HC in Figure 4 to build a picture of the skill of the CRTL-HC predictions made without sea ice thickness initialisation. Figure 4 illustrates that, throughout the CTRL-HC experiment, the seasonal predictions of sea ice extent are consistently biased low. The mean extent over the full
time series (1992-2015) of 4.20 x$10^6$ km$^2$ is between 1.53-2.21 x$10^6$ km$^2$ below that for the 3 observational datasets.

The total extent comparisons in Figure 4a show that the run with initialised winter thickness gives improved predictions of September sea ice extent. This is particularly true for 2011 and 2012, for which the ThkDA-HC predictions of total extent are

within 0.12 x10$^2$ km$^2$ of the observed values. The underestimation of basin-wide extent seen throughout the CTRL-HC predictions has been reduced; 2011-2015 5-year-mean extent of 3.78 x10$^6$ km$^2$ for ThkDA-HC is much closer to the observational average of 4.62 x10$^6$ km$^2$ than is the CTRL-HC value of 2.79 x10$^6$ km$^2$ (Figure 4a).

Basin-wide extent is not a very useful metric for assessing sea ice because, although it provides information about the amount of ice present, it does not take into account the location of the ice or the position of the ice edge - which are more useful for operational users (Notz, 2014). To assess the skill of GloSea seasonal predictions in relation to the spatial distribution of ice and ice edge location, we use the Integrated Ice Edge Error (IIEE) metric introduced by Goessling et al. (2016). This metric is essentially the area integral of all model grid cells where the forecast and observations disagree about whether sea ice is present

or not (see Goessling et al., 2016 for more details). Here we use a sea ice concentration threshold of 15% to define whether ice is present or not in any particular grid cell and compare the GloSea seasonal predictions to the CMEMS reanalysis which assimilated the OSI-SAF data. The GloSea and CMEMS products are on the same ORCA025 grid and so comparisons between the two are easy and not degraded by having to remap the data between different grids. Results from the IIEE analysis can be found in Figure 4b, which shows IIEE for each ensemble member of the CTRL-HC and ThkDA-HC GloSea seasonal

predictions (as in Figure 4a, but for IIEE not extent). The IIEE is virtually flat across the length of the full time series (1992-2015) illustrating that, as for extent, the model without sea ice thickness assimilation is consistently biased throughout this 24-year period.

Figure 4b shows that ice-edge error is considerably improved by the CS2 thickness initialisation with the 2011-2015 mean

IIEE reduced from 3.20 x10$^6$ km$^2$ for CTRL-HC to 2.02 x10$^6$ km$^2$ for ThkDA-HC – a reduction of 37%. The differences in both extent and IIEE shown in Figure 4 are significant at the 1% level over the whole 5-year period and for each of the individual years except for 2013. In general, the improvement in the ice edge location and IIEE is more pronounced than the improvement to the basin-wide extent. This is to be expected given that the CS2 thickness initialisation changed the distribution of sea ice thickness in the Arctic as well as increasing average thickness. Figure 5 further illustrates the spatial improvement

in sea ice predictions showing the probability of ice across the CTRL-HC and ThkDA-HC ensembles for each year (2011-2015) with ensemble-mean and observed ice extent (represented by 15% concentration contours) overlain. Here we calculate the probability of ice, at each grid-cell, as the proportion of ensemble members for which the ice concentration is at least 15%. Consistent with the IIEE results in Figure 4b, the ice edge location in Figure 5 for the ThkDA-HC system is much better than for CTRL-HC. In particular, the ThkDA-HC ensemble-mean ice edges for 2011 and 2012 are very close to those produced by

the CMEMS reanalysis. A consistent feature of Figure 5 is that the ice edge along the Atlantic sector of the Arctic is very well defined for the ThkDA predictions and is very close to the CMEMS reanalysis for all years. These improvements are further illustrated in Figure 6, which shows, for several different Arctic Ocean regions, the ice extent predicted by the CTRL and ThkDA experiments, along with the extent from the CMEMS reanalysis and the corresponding IIEE. The predictions made using CS2 initialisation (ThkDA) have lower extent in the Beaufort and Chukchi Seas and higher extent everywhere else. In

Deleted: very close

Deleted: to

Deleted: hindcasts

Deleted: hindcast

Deleted: forecasts

Deleted: 1

Deleted: down

all regions, the ThkDA extent predictions are closer to the CMEMS reanalysis and the corresponding IIEE is lower. Improvements are most notable in the central Arctic region – and particularly the Atlantic sector.

The spatial changes in the September-mean sea ice concentration predictions depicted in Figure 5 match well with the May

mean thickness dipole shown in Figure 3a. A good illustration of this is 2012 for which the extent improvement is much smaller than the IIEE improvement (Figure 4) which is caused by the fact that much of the ice that remains in the CTRL-HC predictions is located in the Beaufort Sea rather than in the Atlantic sector (north of Fram Strait/Svalbard and east of Greenland). Figure 7 further illustrates this point by showing how thickness differences between the CTRL and ThkDA experiments – for both the analysed spring initial conditions and the September-mean seasonal predictions – relate to the eventual predictions of ice

edge. The thickness dipole from the CS2 nudging matches up well with the areas of missing ice in the Atlantic sector and the areas of excess ice in the Beaufort Sea. This suggests a strong relationship, in this model at least, between wintertime thickness biases and the evolution of errors in sea ice concentration through the summer.

## 4.2 Wider impact of Arctic sea ice changes

We now consider how the abovementioned sea ice improvements affect the wider GloSea seasonal September predictions.

With the changes in winter ice thickness, and in the evolution of Arctic ice coverage through the melt season described above, one would expect to see both fast changes to the local Arctic surface boundary layer (Semmler et al, 2016), as well as longer timescale changes to the wider atmospheric circulation. While much of the recent work on large-scale circulation has focused on changes to winter circulation (Koenigk et al., 2016; Vilma, 2014), studies have shown increased Northern European summer (Screen, 2013; Wu et al, 2013) and East Asian summer monsoon precipitation (Guo et al., 2014) in association with reduced

sea ice.

Figure 8 shows the difference between the ThkDA and CTRL predictions of September-mean near-surface air temperature (T2M), mean sea-level pressure (MSLP), and 500 hPa geopotential height (z500). The left-hand panels show the mean difference, over all ensemble members and all years (2011-2015), between the ThkDA predictions and the CTRL predictions.

Meanwhile the right-hand panels show the mean difference in root-mean-square error (RMSE) between the ThkDA predictions and the CTRL predictions. Here RMSE is calculated for each ensemble member relative to the ERA-I atmospheric reanalysis, which are then averaged over all ensemble members and all years (2011-2015) before differencing. Defining the error with respect to individual ensemble members in this manner, as opposed to looking at the ensemble mean error, provides a sufficiently large distribution of values to allow us to test statistical significance – which we do using a Mann-Whitney test

with the null hypothesis that all errors (or differences) are drawn from the same distribution. The resulting error difference fields calculated using this method are qualitatively the same as considering the difference between the RMSE of each ensemble mean relative to ERA-I (not shown), however the errors here will be larger as there will be no cancellation of errors caused by averaging across ensemble members

We first focus on the local temperature changes, for which Figure 8 shows that, owing to the overall increase in Arctic sea ice thickness and extent, the ThkDA predictions show a general cooling of September T2M, which is significant at the 95% level over most of the Arctic Ocean. This cooling improves the model error relative to the ERA-I atmospheric reanalysis over the majority of this area (Figure 8). The exception to this is south of the Fram Strait in ice export regions, where the T2M has become too cool. We hypothesise that this small increase in error is likely due to the model simulating too much sea ice transport south through the Fram Strait. Interestingly, this improvement is also seen over perennially ice covered regions north of Greenland and the Canadian Arctic Archipelago, where significant improvements in air-sea fluxes would not necessarily be expected. On the Pacific side of the Arctic Ocean, where T2M in the ThkDA experiment is higher than for the CTRL experiments, very little improvement (or degradation) of the T2M is seen.

We next consider the longer timescale quasi-equilibrium response (Semmler et al, 2016) to the pressure fields (MSLP and z500). A significant decrease in MSLP and z500 is seen in the ThkDA experiment over the Arctic Ocean with an accompanying increase over Siberia (significantly so for z500), and with small non-significant increases over the North Atlantic and Pacific (Figure 8). This reduction leads to a decrease in error over the Canadian Basin and Greenland, but slightly worse comparison with observations over the Barents Sea and Western Europe. These differences in error however are generally not significant save for a small patch of improved MSLP over the Canadian Arctic Archipelago (Figure 8).

The z500 and MSLP decrease over the Arctic is suggestive of an increase in both the Arctic Oscillation (AO) and North Atlantic Oscillation (NAO) indices. This is consistent with other studies that have linked lower Arctic sea ice coverage with a tendency for a more meridional atmospheric jet (Francis and Vavrus, 2012), along with a tendency toward the negative phase of the NAO (Petoukhov and Semenov, 2010). It is also broadly consistent with the lower Arctic z500 and wave-train nature of pressure anomalies over Eurasia observed in Wu and Zhang (2013) and Screen et al. (2013) for summertime circulation patterns related to above average sea ice areal coverage. However, owing to the small sample of years looked at here, it is doubtful we could establish a link with increased predictive skill of the inter-annual variability of the atmospheric mid-latitude circulation.

### 4.3 Impact of an improved model thickness climatology

The reanalysis comparison performed in Section 3 revealed persistent thickness distribution biases in the model relative to the CS2 derived data, whereby the ice was too thin in the Atlantic sector and too thick in the Pacific sector. As shown previously (Figure 7) these biases align very well with the ice edge errors suggesting a clear relationship between model thickness bias and forecast error. We would therefore like to understand whether the improvements we see in the GloSea seasonal predictions are caused primarily by an improvement to the model's thickness climatology, or whether the inter-annual thickness distribution changes present in the observations are having an impact.

To try to answer this question another ensemble of seasonal predictions was performed, for years 2011-2014 only, using the 2015 sea ice initial conditions each year. This ensemble of predictions is denoted FIXED-IC. We note here that FIXED-IC predictions are not performed for 2015 because they would simply be a duplication of the ThkDA-HC 2015 predictions. The motivation for adopting this approach is to ensure that we have a dynamically self-consistent initial condition for the sea ice

model. Simply averaging the initial conditions for the 5 years would potentially introduce some coupled initialisation shock that could make the results harder to analyse.

The total extent and IIEE relative to the CMEMS reanalysis for FIXED-IC can be found in Figure 4 alongside those for the ThkDA-HC and CTRL-HC ensembles. The 2015 predictions for ThkDA-HC have been replicated and included as part of the

FIXED-IC experiment. As was the case for the ThkDA experiment, the FIXED-IC predictions are much improved compared with the CTRL-HC experiment. The underestimation of Arctic-wide extent and the IIEE are both reduced. The improvement seen in FIXED-IC is similar in magnitude, but a little lower, than that seen with ThkDA-HC. Although results are worse for 2013, the extent and IIEE analysis in Figure 4 shows ThkDA-HC to be better than FIXED-IC in 2011, 2012 and 2014. With only a short 4-year time series however, it is not possible to distinguish between the FIXED-IC and ThkDA-HC runs

statistically.

Interestingly the FIXED-IC predictions show much reduced inter-annual variability when compared to those from the CTRL and ThkDA experiments and the ensemble-mean extents for each year are close. This is interesting, given that Arctic summer sea ice melt is strongly influenced by atmospheric variability (Deser et al., 2000), and suggests that the ensemble size of 24

used here is sufficient to remove atmospheric variability from the ensemble mean. It also suggests that the initial Arctic thickness distribution and/or volume at the start of the melt season exhibits a controlling factor on the evolution of the ice through the melt season and the eventual September mean extent. This latter point is further supported by the fact that an additional ensemble of GloSea seasonal predictions, performed using constant 2015 initial conditions for both the ocean and sea ice components, gave very similar results to that seen in the FIXED-IC experiment (not shown).

**5 Summary and conclusions**

In this study, we have used nudging techniques to test the impact that initialising sea ice thickness using CryoSat-2 (CS2) measurements could have on Met Office seasonal predictions of September sea ice extent. We have shown that initialisation of sea ice thickness significantly improves the accuracy of GloSea seasonal predictions of summer sea ice cover. Biases in total Arctic extent are reduced as a whole and there are considerable improvements to the spatial distribution of sea ice and

ice-edge location – particularly in the Atlantic sector. These improvements to the sea ice cover also lead to improvements in near-surface temperature and pressure fields over the Arctic domain.

**Deleted:** hindcasts
**Deleted:** hindcasts
**Deleted:** CLIM-2015
**Deleted:** CLIM-2015
**Deleted:** hindcasts
**Deleted:** hindcasts
**Deleted:** doing things this way

**Deleted:** CLIM-2015
**Deleted:** hindcasts
**Deleted:** CLIM-2015
**Deleted:** CLIM-2015
**Deleted:** hindcasts
**Deleted:** CLIM-2015
**Deleted:** CLIM-2015
**Deleted:** CLIM-2015

**Deleted:** CLIM-2015
**Deleted:** hindcasts

**Deleted:** forecasts
**Deleted:** CLIM-2015

**Deleted:** forecasts
**Deleted:** thickness
**Deleted:** forecasts
**Deleted:** forecasts

Technically the application of thickness increments within the CICE sea ice model has been shown to work well. The winter thickness initial conditions, generated using the sea ice thickness nudging, are much closer to the CS2 thickness observations, and lead to considerable improvement in skill when used to initialise GloSea seasonal predictions. The model is able to retain the information supplied by the thickness nudging all the way through the summer when thickness observations are absent.

This is true during the GloSea coupled seasonal forecasts but also for the FOAM reanalysis in which the sea ice model is also being modified by the assimilation of concentration. This result, which is also supported by the findings of Allard et al. (2018), increases our confidence that assimilating sea ice thickness using a more sophisticated and consistent approach will lead to improvements in the FOAM analyses as well as the short-range (FOAM) and seasonal (GloSea) predictions initiated from them.

The motivation for using a simple assimilation approach in this study, using monthly gridded CS2 observations and a nudging technique (as outlined in Section 3), is that it provides a relatively simple way for us to test our hypothesis – that CS2 thickness initialisation will improve seasonal predictions of September Arctic sea ice. The results of this study, made using this approach, suggest that sea ice thickness assimilation within the FOAM ocean-sea ice analysis is feasible and could have a positive impact

on the skill of GloSea seasonal predictions. Motivated by the findings of this study, work is now underway at the Met Office, under the EU-SEDNA project ("Safe maritime operations under extreme conditions: the Arctic case"), to include sea ice thickness assimilation within the NEMOVAR 3D-Var FGAT scheme used in FOAM (Waters et al., 2015), in combination with the sea-ice concentration assimilation already in place. This work will require prescription of observational errors (including instrument, algorithm, and representativeness errors), and the development of methods to represent appropriate

model background errors. It will also involve using raw (L2) satellite tracks, from as many observational platforms as possible (including both CS2 and SMOS), with information being spread through the model using spatial and inter-variable error correlations.

Confronting the sea ice thickness from the FOAM reanalysis with the CS2 satellite data has revealed a persistent bias in the

modelled thickness distribution whereby the simulated Arctic sea ice is too thin on the Atlantic side and too thick in the Beaufort Sea. This bias is most likely caused by deficiencies in the formulation of the sea ice dynamics: either the rheology, or deficiencies in the momentum exchange between components in the atmosphere-ice-ocean (primarily wind drag). To ameliorate this situation we plan to experiment with the form-drag scheme and the anisotropic rheology developed for CICE by the CPOM group at University of Reading (Tsamados et al., 2013; 2014). In particular, the form-drag scheme has been

shown to improve the Arctic thickness distribution in standalone sea ice model experiments (D. Schroeder, pers. comm., 2017).

The clear relationship between modelled winter thickness biases and summer extent errors shown in Figure 7, along with the improved ice cover obtained using thickness initialisation (Figure 4 and Figure 5), highlights the importance that modelled winter thickness biases can have on the evolution of forecast errors through the melt season. Results from the FIXED-IC

experiment further suggest that the ensemble size of 24 used here is sufficient to account for atmospheric variability and that late winter/spring sea ice thickness provides quite a strong constraint on the eventual September extent (Figure 4) in any particular year.

Although the addition of sea ice thickness nudging to the FOAM analysis system clearly improves the seasonal predictions of summer sea ice, it is not clear how much of this improvement comes from initialising each year with the CS2 thickness and how much is due to the assimilation improving the model's thickness distribution climatology. The IIEE and extent analysis suggests that, for 3 out of the 4 years, using the correct thickness initialisation (ThkDA-HC) provides a better prediction of September ice edge location when compared with the run using the 2015 thickness (FIXED-IC). This is in agreement with the

findings of Day et al. (2014) who showed that, in the perfect model framework, that correct initialisation of Arctic thickness in the HadGEM1 climate model, led to an improved model evolution when compared with initialising the model with its own thickness climatology. In this case, however, we are not able to say this conclusively because the time series is too short to allow us to reject the null hypothesis that all the ensemble members from these two runs are taken from the same distribution.

Furthermore, the improvement shown in Figure 4 between ThkDA and FIXED-IC is small relative to the improvement between ThkDA and CTRL. Therefore, we conclude that, certainly for the GloSea seasonal prediction system, improving the model thickness climatology is at least as important as initialisation of sea ice thickness for improving predictive skill of seasonal forecasts.

*Acknowledgements. We acknowledge funding support from: the Joint UK BEIS/Defra Met Office Hadley Centre Climate Programme (GA01 101); the European Union's Horizon 2020 Research & Innovation programme through grant agreement No. 727862 (APPLICATE); the UK-China Research and Innovation Partnership Fund through the Met Office Climate Science for Service Partnership (CSSP) China as part of the Newton Fund; and the UK Public Weather Service research programme. We are thankful to Philip Davis (Met Office) for running and providing access to the CTRL-HC prediction ensemble. Provision*

*of observational data sources used within this study is also acknowledged: CryoSat2-derived thickness fields from CPOM; sea ice concentration products from OSI-SAF and NSIDC; ERA-Interim atmospheric reanalysis from ECMWF. EB would further like to thank Andy Ridout (CPOM, UCL) for providing monthly CS2 data and for his useful advice regarding aspects of the quality control.*

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

**Table 1: Details of ocean-sea ice reanalysis experiments used in this study**

| Experiment name | Renalaysis run period | Surface forcing | Assimilated variables (3D-Var) | Thickness nudging used |
|---|---|---|---|---|
| CTRL-RA | 01/01/1992 – 31/12/2015 | ERA-Interim | SST, SLA, T&S, SIC | None |
| ThkDA-RA | 01/10/2010 – 31/12/2015 | ERA-Interim | SST, SLA, T&S, SIC | CPOM CryoSat-2 |

Deleted: R

Deleted: Reanalysis

5 **Table 2: Details of GloSea coupled seasonal prediction experiments (or 'hindcasts') used in this study**

Deleted: hindcast

| Experiment name | Prediction lead time | Years | Ensemble members | Atmosphere ICs | Ocean/Ice ICs |
|---|---|---|---|---|---|
| CTRL-HC | May – Sep | 1992-2015 | 24 per year[a] | ERA-Interim | CTRL-RA |
| ThkDA-HC | May – Sep | 2011-2015 | 24 per year[a] | ERA-Interim | ThkDA-RA |
| FIXED-IC | May – Sep | 2011-2014 | 24 per year[a] | ERA-Interim | ThkDA-RA: fixed 2015 for all years |

Deleted: Hindcasts

Deleted: Run period

Deleted: CLIM-2015

[a] 24 members per year = 3 start dates with 8 stochastic members for each

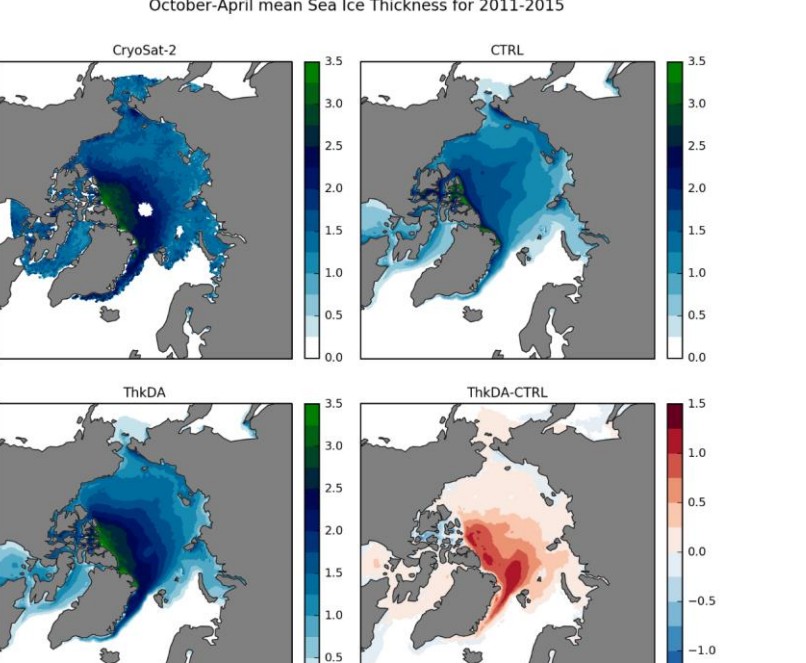

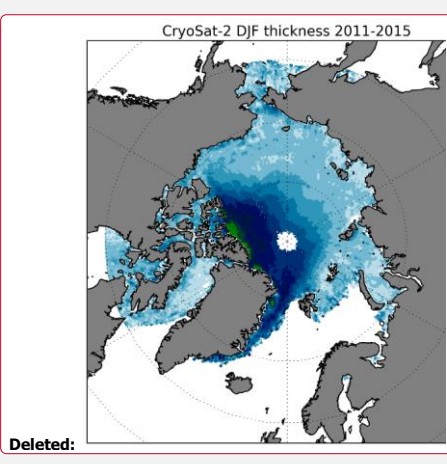

Figure 1: **Mean winter (October to April) Arctic sea ice thickness (m) from October 2010 to April 2015. Showing data from the CPOM CryoSat-2 measurements (after application of the QC and imposing the 1m minimum thickness threshold), along with modelled thickness from the CTRL and ThkDA reanalyses. The lower-right panel shows the difference between the ThkDA and CTRL experiments.**

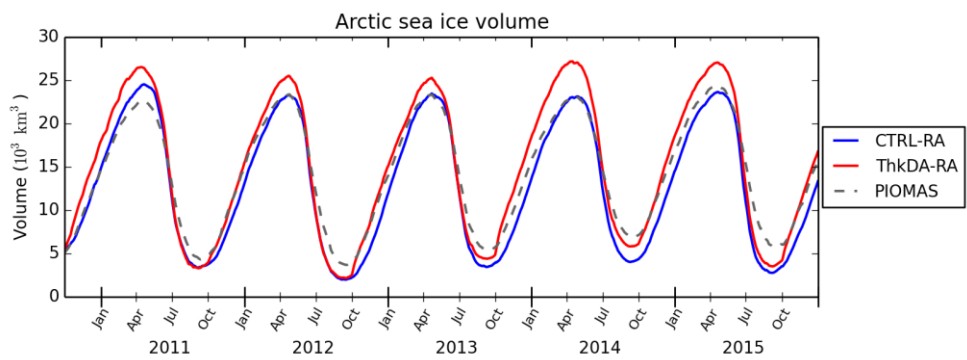

**Figure 2**: Reanalysis Arctic sea ice volume from 1st October 2010 to the end of December 2015 from the CTRL-RA (blue) and ThkDA-RA (red) reanalysis experiments. Sea ice volume from the PIOMAS model (grey dashed) is included as a reference.

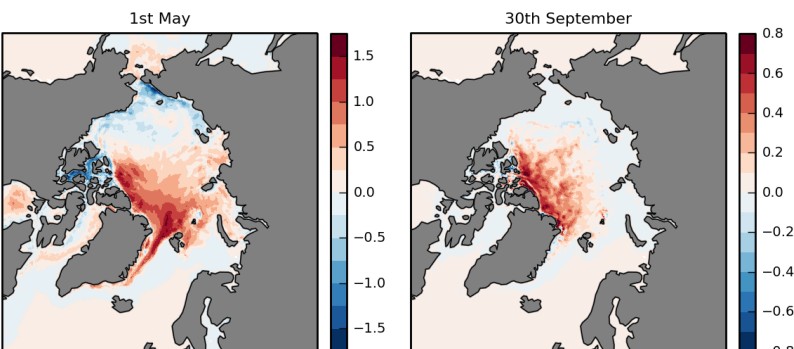

**Figure 3**: Mean sea ice thickness difference (m) between ThkDA-RA and CTRL-RA experiments over the full 5-year reanalysis period from 2011 to 2015. Showing differences for (left) the end-of-winter on 1st May used for the initialisation of summer seasonal forecasts, and for (right) the end-of-summer on 30th September. The difference is taken as ThkDA – CTRL so red (blue) implies that the CS2 nudging has increased (decreased) the thickness.

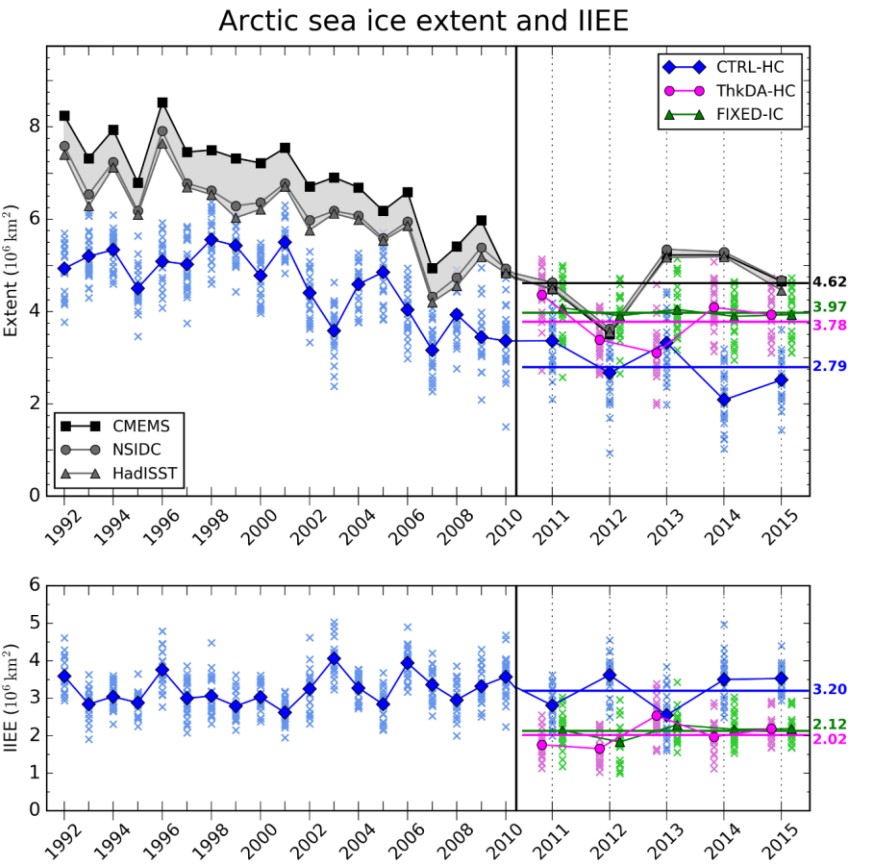

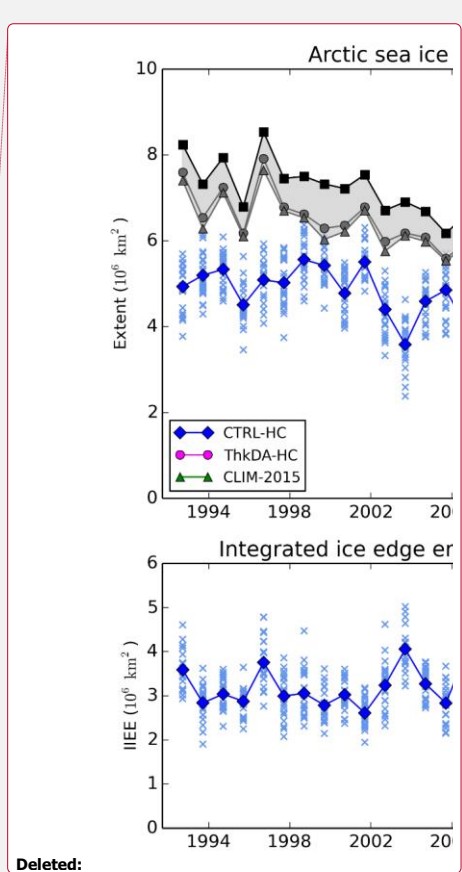

Figure 4: (upper) September-mean Arctic sea ice extent from the CTRL-HC (blue), ThkDA-HC (pink) and FIXED-IC (green) seasonal prediction experiments. Observational estimates from the CMEMS reanalysis assimilating OSI-SAF (black square), NSIDC (grey circles) and HadISST1.2 (grey triangles) are included and the area between them shaded light grey. (lower) Integrated Ice Edge Error (IIEE) for seasonal predictions relative to the CMEMS reanalysis product introduced in Section 2.3. In both panels, individual ensemble members are represented by coloured crosses and ensemble means by the solid symbols and inter-connecting lines. Horizontal coloured lines depict 2011-15 mean values. For ease of viewing, the ThkDA-HC (pink) and FIXED-IC (green) experiments are plotted with a small offset relative to the CTRL-HC (blue) experiment, and the CS2 period (2011-2015) is plotted with an increased x-axis scale, approximately twice that for the early period 1992-2010, with the transition indicated by a vertical black line.

September forecast probability of ice (conc>15%)

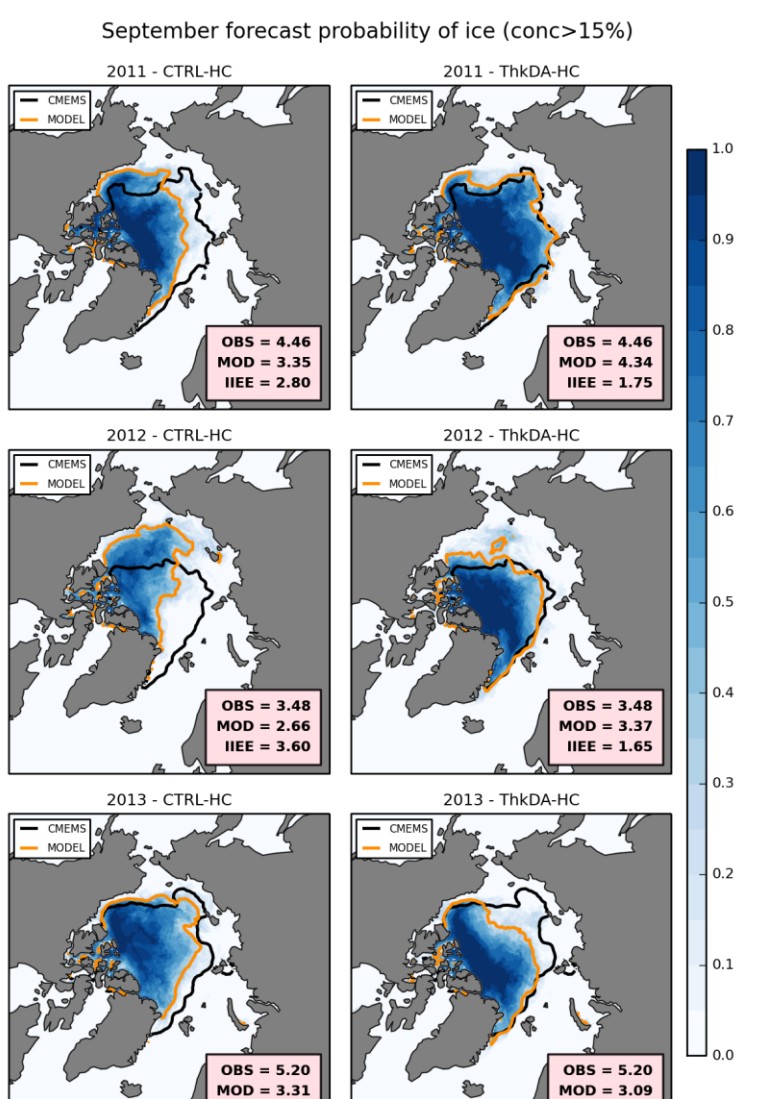

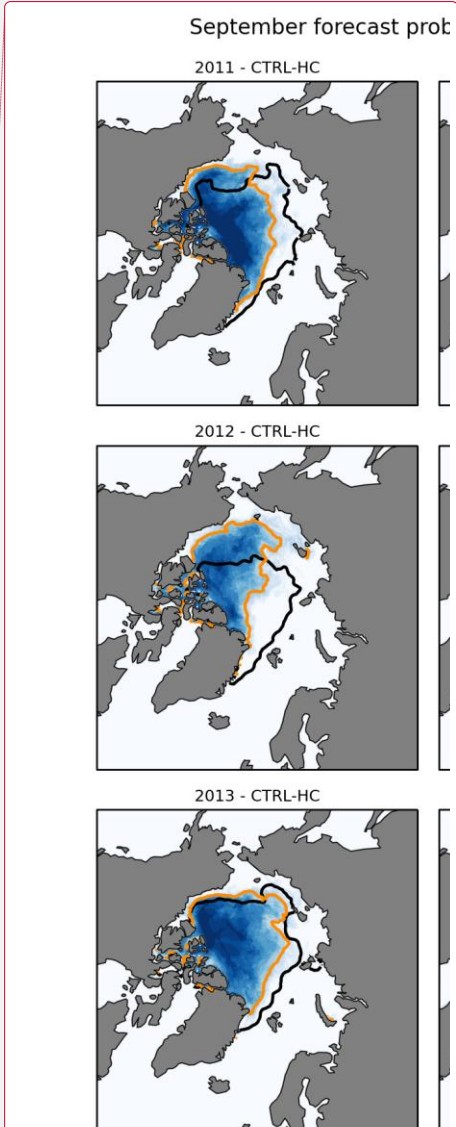

September forecast prob

**Deleted:**

## September forecast probability of ice (conc>15%)

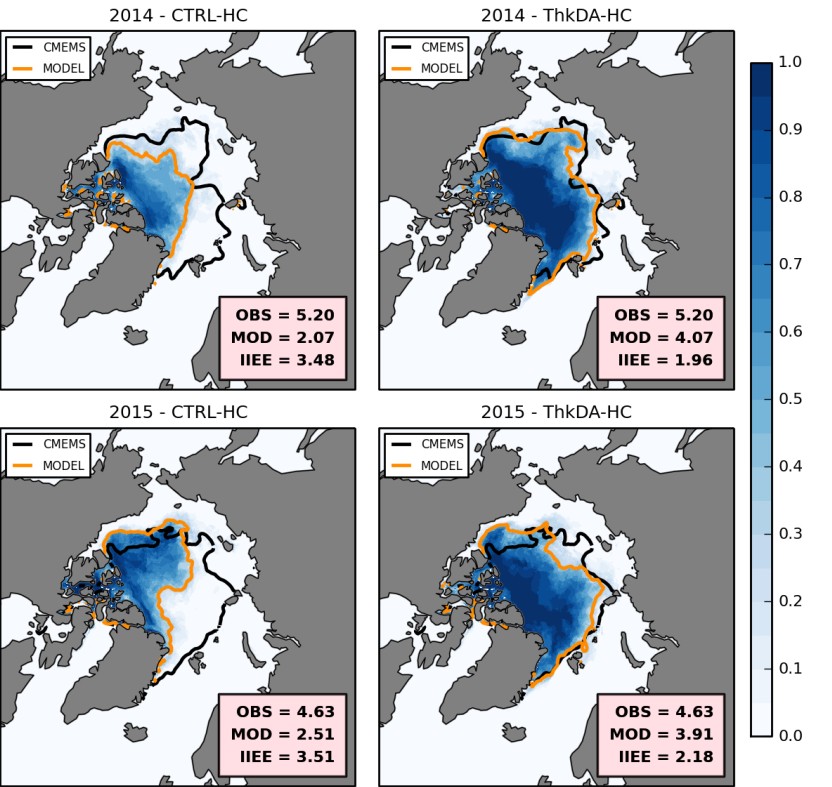

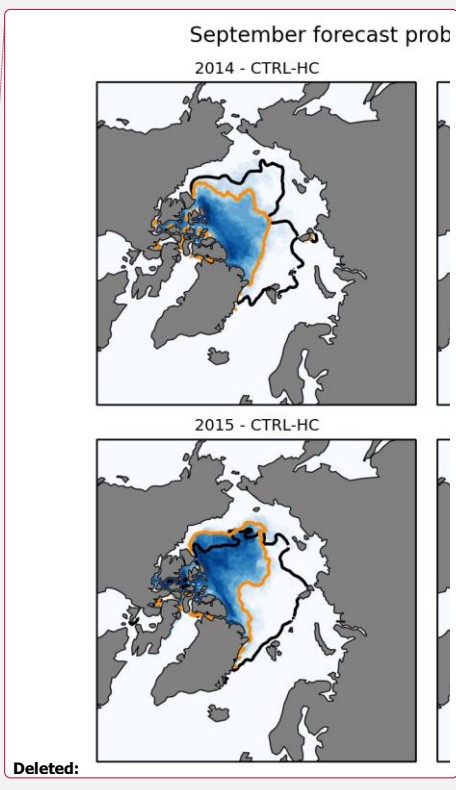

**Figure 5: September mean probability of sea ice for the CTRL-HC (left) and ThkDA-HC (right) seasonal predictions for all years from 2011 (top) to 2015 (bottom). Contours of 15% concentration are overlain to represent the sea ice edge for the ensemble mean (orange) and CMEMS reanalysis product (black). Probability is defined at each point as the proportion of ensemble members that have at least 15% ice concentration. The CMEMS extent, modelled extent and corresponding Integrated Ice Edge Error (IIEE) are included, for each plot, in the lower-right corner (units: $10^6$ km$^2$).**

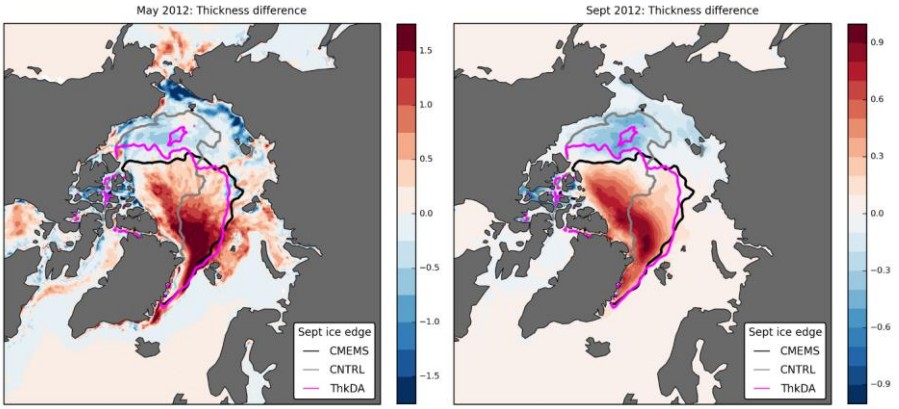

**Figure 6: Mean September Arctic sea ice extent for 2011-2015 from the CMEMS reanalysis (using OSI-SAF) compared with modelled extent and ice edge error (IIEE) from the CTRL and ThkDA seasonal predictions (units x10⁶ km²). Data are shown for 3 regions distinguished by the underlying shading and corresponding box colours: combined Beaufort + Chukchi Seas (yellow), combined Kara, Laptev and East Siberian Seas (dark blue) and the central Arctic (red). Also shown (pink boxes) are corresponding statistics for the Atlantic and Pacific sectors of the Arctic Ocean, defined by splitting the Arctic Ocean (i.e., red + yellow + dark blue) along 30°W and 140°E longitude (yellow lines) – which roughly follows the Lomonosov Ridge.**

**Figure 7: Thickness difference (m) between ThkDA and CTRL experiments in May and September 2012 (contour shading) with differences calculated as ThkDA – CTRL in each case. (left) shows the difference between the reanalyses fields used to initialise the seasonal predictions on 1st May 2012. (right) shows ensemble mean forecast differences for September 2012. (Note the different scales used for the coloured shading). Overlain on both panels are September-mean contours of 15% ice concentration to represent the sea ice edge for the CTRL-HC (grey) and ThkDA-HC (pink) experiments along with the CMEMS reanalysis product (black).**

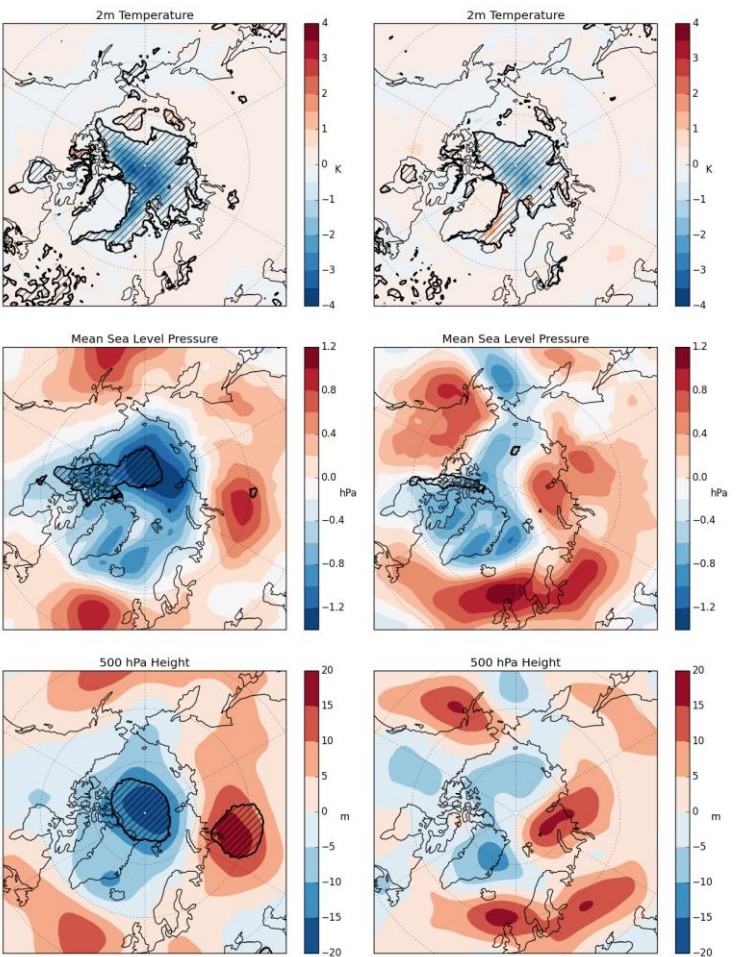

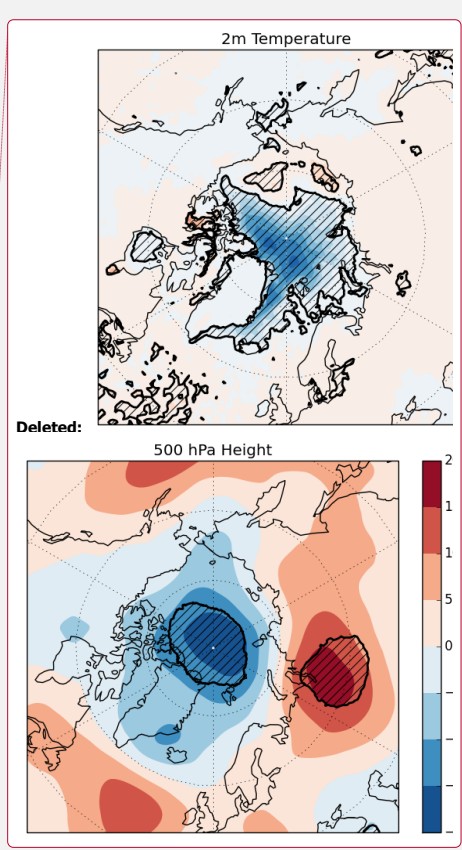

**Figure 8: Differences in mean fields (left) and root-mean-square-error (RMSE; right) between the ThkDA and CTRL** September predictions averaged over all ensemble members for the 5-year period 2011-2015. Fields shown are: (top) near-surface 2m air temperature (T2M; K); (middle) mean sea-level-pressure (MSLP: hPa); (bottom) 500 hPa geopotential height (z500; m). Differences
5    are calculated as ThkDA – CTRL meaning that areas of blue (red) denote that ThkDA predictions/RMSEs are lower (higher). Black contours and hatching denote areas where differences are significant at the 95% level as determined using a Mann-Whitney U-test. Further details can be found at the beginning of Section 4.2.

