# Peer review of "Improving Met Office seasonal predictions of Arctic sea ice using assimilation of CryoSat-2 thickness"

_The Cryosphere, 2018_

## Referee Comment (RC1) · Anonymous Referee #1 · 12 May 2018

This paper explores the benefits of assimilating winter sea ice thickness estimates from CryoSat-2 into the Met Office FOAM/GloSea sea ice assimilation and seasonal forecast system. The approach builds on recent work demonstrating the utility of sea ice thickness assimilation in perfect model and forced model simulation studies. The authors briefly explore the atmospheric response to an improved representation of summer sea ice.

The approach taken was simple - nudging the simulated winter sea ice thickness towards the CryoSat-2 observations and assessing the impact on the simulated ice state. Much of this was unsurprising - the simple nudging scheme worked and reduced a

modeled thickness bias. The analysis of the potentially more interesting benefits of assimilating sea ice thickness were sometimes less clear, including the general aims of the paper beyond demonstrating this thickness bias reduction.

I think the paper could be worth publishing if several key improvements are made as detailed below. Note that I am not an expert in data assimilation so hopefully another reviewer can provide more input regarding the approach taken and their method of evaluation.

Also a general gripe of this study and similar ones: I find it really challenging to understand what you mean by the use of forecast and hindcast throughout the paper and they seem to be used interchangeably. If you are using prescribed atmospheric forcings that have assimilated real data then my view is anything using that is a hindcast not a forecast. Prediction is the more general term that could be appropriate but I see nothing in this paper that resembles a true forecast (no future knowledge), despite the title.

Main comments:

It's really not clear from the motivation what it is you are trying to achieve by assimilating sea ice thickness in this study. In some cases you say the ice area/extent impact is negligible (as you ignore lower thickness ice) but other times it seems you highlight big improvements in your ice edge 'forecast'. You should really present a hypothesis you are testing in this kind of study (i.e. which metrics you are assessing). In general I would think assimilating thin ice should be especially important for seasonal (spring/summer) forecasting as you want to correctly incorporate this into your model as this is the ice most likely to melt out. If you get that wrong, you get the summer melt wrong. It seems like you used the opposite reasoning to justify not using the AWI/SMOS data but your results suggest the opposite to be true if you care about both the summer ice thickness and ice edge.

I had a lot of issues with Section 2.2 (describing the CryoSat-2 data): - You need to

make clearer the various thickness datasets available and how a number of groups are now routinely producing sea ice thickness estimates, e.g. NASA/AWI/CPOM for CryoSat-2. - CPOM and NASA data were used in the study of Allard 2018. You need to make this clearer. It was in general unclear how your study differed from Allard and I think you need more discussion of their approach and results. - Is the pole hole and the data uncertainty really why these data haven't been used? It seems like any reasonable assimilation scheme shouldn't need complete coverage and can factor in data uncertainties. My guess is that the main reason was data availability, the fact these thickness products were in their infancy, and inertia. - On that note, how do you treat the fact you are unlikely to have complete coverage from the CryoSat-2 data? - After listing problems associated with generic thickness data you then say this is improved by the availability of CS-2 thickness data. This doesn't make much sense as it is written. - You should cite the relevant studies regarding uncertainty estimates, not just the Ricker/AWI reference and apply that to CS-2 data derived by other groups. There are strong differences in the retracking procedures which may have impacts on respective data uncertainties across the products. - I don't get what this extra quality checking of the CPOM CS-2 data is. The fact you have included a personal comms from one of the data producers of that dataset makes it seem like this is something they do too? What exactly do you mean by smearing?

Why not use daily along-track CryoSat-2 data? I thought this was the whole purpose of CPOM releasing the daily along-track data? Instead they grid the data, then you grid the data, then you interpolate to a get a daily thickness?

Hard to tell what this volume comparison really means. You compare with PIOMAS but then say that data is biased low (which I'm not actually sure is true when you look at more than one CS-2 estimate) so it is actually good that you are further from that data? You say this was expected but this seems like a hindsight statement to me. I agree PIOMAS data can provide useful context but I don't agree with how you've used it. I think you should just show the CS-2 data and say look, the assimilation does what it is

supposed to do..

P10 L10-20 and elsewhere: Very confusing to me if these are hindcasts or genuine forecasts. You use both labels interchangeably. How could you move forward to produce genuine forecasts?

Specific comments:

In the abstract: really the 'first time, we directly assess the impact of winter sea ice thickness initialisation on the skill of seasonal summer forecasts'? Do you mean in the Met Office model framework? I think Allard and others have done this and I also don't think you do assess forecast skill

Introduction - In general you need more updated references. A lot of this discussion is a bit outdated now. i.e.:

- Drop the Vaughan/IPCC refs and use the more specific refs. Try Serezze & Stroeve 2015 for a more recent seasonal sea ice trends citation?

- The Collins/IPCC is also a bit outdated. I think you can add some of the more recent references to sea ice projections - e.g. Jahn 2016, 2018.

P2 L1-11 - I think this is not useful information as it is not that relevant to seasonal forecasting and a lot of the references and discussion are outdated. Either update/improve or drop.

P2 L13 - change sentence ordering.

P2 L17 - the predictive skill sentence is confusing. SIPN haven't really assessed that.

P2 L19-23 - I don't think you've really said why it is interesting though! Either make a clearer point regarding its scientific interest (e.g. what the predictability/memory of the system is compared to other components of the climate system). Also I don't think it is clear that sea ice is now necessarily harder to predict. Having some enhanced variability may be useful. Your figure 4 doesn't show an increase in ice edge error for

instance!

P2 L27 - why exactly does a lack of observations make the forecasts harder? Less to assimilate in models or to validate? Below you list a number of observations that are available in the poles...

P2 L35 - include acronym definitions.

P3 L4 onwards - this is a bit of a confusing paragraph to me. What is the point you are trying to make? In general my view is that there is hope for dynamical models being used for skillful sea ice forecasting based on some of the perfect model studies that you cited. However the SIO has really shown that they are not currently performing much better than the linear trend in many cases (as shown by Stroeve et al., 2014). It still seems that dynamical models are lagging behind more simple statistical methods (e.g. Schroeder et al., 2014, Petty et al., 2017) despite their sophistication.

I think you need to add in some comments on the different forecast methods available, merge with the following paragraph about improving dynamical models and make clearer what the motivation of this study is! This should be the key paragraph of the introduction.

P3 l 24 - not sure how this point links to the above.

P4 L1-4 - but in the abstract you imply you are the first to do this?! I guess you meant in your fully coupled Met Office forecast framework. You need to make that clearer.

P5 L11-13 - why just mention the ocean reanalysis component here? Would be more understandable if you referred to GloSea as a reanalysis.

P5 L23 - I don't think these are the correct citations here. Link to relevant passive microwave concentration datasets instead or recent papers describing that long-term record (e.g. Parkinson/Comiso papers).

P5 L25 - should reference Kwok and Cunningham 2008 instead.

P6 L16-17 - what do you mean by sensitive here? I think you mean uncertain/challenging? Again, is there nothing in one of the CPOM papers that highlights this issue?

P6 L30 - I think this is a big guess. Do we really know much ice < 2 m melts away each summer? That Keen modeling study (Fig 2?) suggests some 40% of the ice less than 2 m (including ice and snow in that thickness) does not melt through in summer. A lot will have to do with how much snow there was on the ice and where/when melt onset occurs. Even if it does melt away, this seems to be crucial information for determining solar absorption that can drive SST increases and further sea ice melt. Also need to make the point here (and earlier) that AWI do produce a merged product!

Section 2.3 - Why do you need extent and concentration? Surely you are just assimilating sea ice concentration? This needs to be made much clearer here. - The use of CMEMS 'data' seems very confusing to me. Why not use observed ice concentration?! - The NSIDC sea ice index is just a monthly index of total ice extent. This isn't what you use, right?

P8 L6 - what is the size of your model grid (in kilometers?) how does this translate to the 5 km CPOM data?

P8 L14-24 - Think you should list out the CICE thickness categories. How else could you have done this? I think it would be worth presenting more sophisticate approaches for future work, however I get that you started with this simple approach.

Figure 2 - I think you should also show (maybe in the supplementary info?) what the pre nudged, and nudged thickness fields are. Could just do this for the mean October-April thickness and also update Figure 1 to show this longer season too. If the mean thickness was way off before it makes sense that assimilating the thickness will improve things..

Why the different start dates for the forecasts? Pretty confusing.
Figure 4 is hard to see. Maybe box plots of the recent years showing the variability in the different estimates in the different years?

P11 L10 - CMEMS isn't really an observational estimate, right? Based on the assimilation of OSI-SAF...

Pll L14 - not sure what you mean by 'building a picture' here. I see no value in showing that earlier data.

P11 L17 what is close

Figure 5 - include numbers on map. Plot the IEEE as a time series.

Figure 7 and 8 - I don't understand these maps. How exactly is the data shown in Figure 8 calculated? Also is Figure 8 averaged over the entire year, but figure 7 is September? Why are the pressure units different? Perhaps better to show Figure 7a and 8a together, then 8a and 8b.

The discussion of using ensemble members instead of the ensemble means and how this relates to assessing model bias was confusing and needs more description/clarification.

Should either enhance the CLIM analysis or drop. i.e. repeat with using a fixed (think FIXED_IC would be a better acronym) for all years of data available.

How do these results compare with Allard?

P14 L21 - not sure what you mean by work well. Maybe not gone wrong?

P14 L1 Why is this from dynamics not thermodynamics or the forcing being off?! Also drop 'so-called ice-ice force'
* * *

---

## Referee Comment (RC2) · Anonymous Referee #2 · 17 May 2018

General Comments

In this paper, the authors use a version of the "Met Office" fully coupled atmosphere-ocean-sea ice modeling system to examine the impact of the seasonal skill (improvement) of the September sea ice by assimilating "winter" CryoSat-2 ice thickness data from CPOM. A control run without this data is compared to a simulation for the period of 2010-2015 in which CS2 data was assimilated via a nudging method for three specific sets of dates: April 25, May 1, and May 9 with 8 ensemble members each. This study represents the first known fully coupled atmosphere-ocean-ice forecast system to utilize CryoSat-2 ice thickness data for seasonal forecasts. The underestimation of

the basin-wide extent was reduced for this period from 3.78X106 versus the control run (2.79x106) versus the observation average of 4.62x106. Another study is performed to examine ice edge error by using an Integrated Ice Edge Error (IIEE) metric in which a 37% reduction was observed with the CS2 initialization experiment. A volume comparison between the Control and CS2 runs shows an increased volume with the CS2 initialization; the PIOMAS volume is consistently lower except for the summer months. Some improvement was found in the 2-m air temperature in September, where the error was determined as the difference between the model and the ERA-I reanalysis. Differences were also noted for the 500hPa thickness and sea level pressure; but with the limited time period examined (5 years), the authors could not make a direct connection to the AO and NAO.

Overall, this study shows promising results. I recommend publication with minor revisions. See comments below.

Specific Comments

Page 3 Line 1: Add Cummings and Smedstad (2014) for another coupled ocean-ice modeling system with data assimilation, here is a full reference: Cummings, J. A. and O. M. Smedstad, 2014: Ocean Data Impacts in Global HYCOM, Journal of Atmospheric and Oceanic Technology, 31, doi:10.1175/JTECH-D-14-00011.1

Page 7 line 31: why couldn't a longer period, say from 2010-2017 or at least 2010-2016 be used?

Page 8 line 14: which version of CICE (v4.0, 4.1) is used. Are melt ponds used in this study?

Page 14 line 17-18: can you quantify bias reduction with some range or percentages?

Figure 4: Please add an inset for both plots showing a blow-up for the period 2010-2015? It's a bit difficult to see with the longer data record shown.

There is no mention of ice drift in the paper. Could you analyze IABP ice drift data

(Pan-Arctic domain) to determine the impact of assimilating CS2 data into the seasonal forecasts? This would complement your existing study. SIDFex is presently examining modeling center's skill in making long-term ice drift trajectory forecasts.

I would like to see the ice edge error metric used to examine the regional differences seen from use of the CS2 data. Can it be divided into the following (or similar) basins (Beaufort/Chukchi/Bering Sea, Canadian Archipelago, Greenland Sea, Laptev Sea, Barents Sea, East Siberian Sea)?

No comparisons are made against ice thickness observations from either ice mass balance buoys and/or moored ULS data. I recommend inclusion of some time series plots of the modeled ice thickness beginning with the Apr/May initializations through September for 2010-2015, with the control run included. The ensemble spread can be shown as well. This should clearly show the impact of the inclusion of the CS2 data.

Lots of acronyms are used without spelling them out. A partial list is shown below. Perhaps a list or table of acronyms would be useful.

Technical Corrections Page 2 line 6: replace "Better knowledge" with "Improved knowledge"

Page 2 line 34: define SLA here

Page 3 line 22: define CFSv2

Page 3 line 25: replace "find" with "found"

Page 3 Line 34: Yang et al. reference not listed in References

Page 4 line 3: spell out NRL (Naval Research Laboratory)

Page 5 line 3: spell out FGAT

Page 5 lines 12-14: "used" appears in sentence 3 times. Perhaps change second mention of this word to "utilized"

Page 5 line 26: reword statement "have been around for some years"

Page 5 line 27: replace "main" with "primary"

Page 5 line 32: spell out SIRAL

Page 6 line 22: replace "was" to "is"

Page 7 line 16: spell out SSMI/S

Page 7 line 28: Ridley reference says (2017, in review); reference section states 2018

Page 9 line 24: reword to "although IT has".

Page 10 line 4: delete "the fact"

Page 10 line 7: reword "amongst other things" what things?

Page 10 line17: Williams reference in references section says 2018

Page 10 line 25: MacLachlan reference says 2014 in references section

Page 12 line 2: delete "down"

Page 13 line 26: reword "doing things this way"

Page 14 line 28: spell out SEDNA

Page 15 line 4: Neither Tsamodos reference is listed in References section

Page 15 line 17: replace "down" with "due"

Page 21: Peterson reference should be 2015 not 2014

---

## Author Comment (AC1) · 3 Aug 2018

**Response to review #2**

We thank the referee for their review of our manuscript. Answers to the specific comments in the review are listed inline below. The review comments are in black italics and our response is in red.

**Specific Comments**

*Page 3 Line 1: Add Cummings and Smedstad (2014) for another coupled ocean-ice modeling system with data assimilation, here is a full reference: Cummings, J. A. and O. M. Smedstad, 2014: Ocean Data Impacts in Global HYCOM, Journal of Atmospheric and Oceanic Technology, 31, doi:10.1175/JTECH-D-14-00011.1*

The articles cited here (i.e., Tonani et al., Martin et al., Balmaseda et al.) are all large multi-centre papers which describe all of the global ocean/ice analysis + forecasting systems that contribute to the GODAE OceanView project. This includes HYCOM-based systems such as that documented in Cummings and Smedstad (2014). Jim Cummings is part of the author list for one of these, and several Cummings papers are cited. However, this exact paper is not one of those referenced in the papers we cite here and so we shall add this citation to the revised manuscript.

*Page 7 line 31: why couldn't a longer period, say from 2010-2017 or at least 2010-2016 be used?*

This work was started in 2016 and so the analysis was performed up to September 2015, which was the most recent summer period at that time. This is true for all the control runs that we used in this project as well as the CS2 runs that we performed ourselves. Whilst undertaking this work, we were only able to perform such a large ensemble of seasonal predictions because the delay finalising our model configuration for CMIP6 meant that there was a good chunk of computational resource available. This is no longer the case and CMIP6 runs are in full swing now – meaning that we do not have the computational resources available to extend this to 2016 or 2017 at this time.

*Page 8 line 14: which version of CICE (v4.0, 4.1) is used. Are melt ponds used in this study?*

CICE vn5.1.2 is used here and the configuration includes topographic melt ponds. The coupled model version used here (GC3) is documented in Williams et al. (2017) and the sea ice component therein (GSI8) is documented in more detail in Ridley et al. (2018). Therefore we do not intend to include lots of technical information about the sea ice model used here.

However we agree with the referee that more information is required here and so we shall expand the text at the end of Section 2.1 to include key features of the sea ice model component (such as multi-layer thermodynamics, 5 thickness categories, prognostic melt-ponds, etc.).

*Page 14 line 17-18: can you quantify bias reduction with some range or percentages?*

Yes we will do this in the revised manuscript. This will take the form of quantifying mean SIT differences – as a value or a percentage –within the Atlantic and Pacific sides of the Arctic. We shall also extend Figure 2 to include plots of mean thickness for the control and thkDA runs to help illustrate this.

However, we feel that this would sit better at the end of page 9 where the bias/difference is first discussed rather than on page 14.

*Figure 4: Please add an inset for both plots showing a blow-up for the period 2010-2015? It's a bit difficult to see with the longer data record shown.*

Reformatting Figure 4 was also requested by referee #1. We are trialling different options for this at present – using insets and/or additional panels. The revised manuscript will include an improved Figure 4 that allows the results to be more clearly displayed. More space on the plot will be devoted to the key time-period of 2011-2015, and we will ensure that the individual ensemble member values are more easily distinguished.

*There is no mention of ice drift in the paper. Could you analyze IABP ice drift data (Pan-Arctic domain) to determine the impact of assimilating CS2 data into the seasonal forecasts? This would complement your existing study. SIDFex is presently examining modeling center's skill in making long-term ice drift trajectory forecasts.*

We have compared the May and September sea ice velocity fields from our two main experiments (CTRL and thkDA). We find that the May velocities are virtually indistinguishable in each of the 5 years (2011-2015). This is consistent with the findings of Allard et al. (2018) who show little impact on ice drift in their reanalysis comparisons.

The September velocity fields are also very similar. Although slight differences arise from the differing ice coverage, the ice drift is broadly/qualitatively unchanged in the experiment using sea ice thickness initialisation.

Thanks for the pointer to SIDFEx, which is something we've been meaning to get involved with for some time. We are in the process of setting things up to contribute to SIDFEx using our FOAM and coupled NWP systems (but sadly not from the GloSea seasonal prediction system because that only outputs monthly-mean ice drift).

*I would like to see the ice edge error metric used to examine the regional differences seen from use of the CS2 data. Can it be divided into the following (or similar) basins (Beaufort/Chukchi/Bering Sea, Canadian Archipelago, Greenland Sea, Laptev Sea, Barents Sea, East Siberian Sea)?*

Good idea. This would be a useful to underline the key points of this work. We have calculated the IIEE for several Arctic regionals and are exploring ways to display this information. This will most likely lead to an additional figure along the lines of Figure 1 (at the end of this document).

These results are consistent with the rest of the results here: runs with CS2 initialisation have decreased extent in the Beaufort & Chukchi Seas and increased extent everywhere else. In both cases, this brings us closer to the observations/reanalysis and lowers the ice-edge error (IIEE). Improvements are most notable in the central Arctic region - and particularly the Atlantic sector.

*No comparisons are made against ice thickness observations from either ice mass balance buoys and/or moored ULS data. I recommend inclusion of some time series plots of the modeled ice thickness beginning with the Apr/May initializations through September for 2010-2015, with the control run included. The ensemble spread can be shown as well. This should clearly show the impact of the inclusion of the CS2 data.*

While comparison with ULS would be interesting, we feel it is out of scope for this study where we are heavily focussed on improvement to sea ice cover (extent and edge location). However, this sort of comparison is something we would wish to do before implementing a proper 3D-Var sea ice thickness assimilation scheme – and in fact is currently being undertaken within the H2020 SEDNA project.

*Lots of acronyms are used without spelling them out. A partial list is shown below. Perhaps a list or table of acronyms would be useful.*

We apologise for this oversight. The revised manuscript will ensure that acronyms are spelled out at the point of first use. We do not think that this manuscript is "acronym heavy" enough to need a glossary appendix/table though. However, we can include one if the referee (or the journal typesetters) feels strongly about this.

**Technical Corrections**

*Page 2 line 6: replace "Better knowledge" with "Improved knowledge"*

*Page 2 line 34: define SLA here*

*Page 3 line 22: define CFSv2*

*Page 3 line 25: replace "find" with "found"*

*Page 3 Line 34: Yang et al. reference not listed in References*

*Page 4 line 3: spell out NRL (Naval Research Laboratory)*

*Page 5 line 3: spell out FGAT*

*Page 5 lines 12-14: "used" appears in sentence 3 times. Perhaps change second mention of this word to "utilized"*

*Page 5 line 26: reword statement "have been around for some years"*

*Page 5 line 27: replace "main" with "primary"*

*Page 5 line 32: spell out SIRAL*

*Page 6 line 22: replace "was" to "is"*

*Page 7 line 16: spell out SSMI/S*

*Page 7 line 28: Ridley reference says (2017, in review); reference section states 2018*

*Page 9 line 24: reword to "although IT has".*

*Page 10 line 4: delete "the fact"*

*Page 10 line 7: reword "amongst other things" what things?*

*Page 10 line17: Williams reference in references section says 2018*

*Page 10 line 25: MacLachlan reference says 2014 in references section*

*Page 12 line 2: delete "down"*

*Page 13 line 26: reword "doing things this way"*

*Page 14 line 28: spell out SEDNA*

*Page 15 line 4: Neither Tsamodos reference is listed in References section*

*Page 15 line 17: replace "down" with "due"*

*Page 21: Peterson reference should be 2015 not 2014*

Many thanks for providing such a thorough list of technical changes. We shall of course incorporate all of these changes in the revised version of the manuscript.

[Figure]

**Pacific sector**

|      | Extent | IIEE |
|------|--------|------|
| OBS  | 2.82   | -    |
| CTRL | 2.12   | 1.63 |
| thkDA| 2.37   | 1.07 |

**Siberian Shelf**

|      | Extent | IIEE |
|------|--------|------|
| OBS  | 0.54   | -    |
| CTRL | 0.24   | 0.57 |
| thkDA| 0.25   | 0.5  |

**Beaufort + Chukchi**

|      | Extent | IIEE |
|------|--------|------|
| OBS  | 0.54   | -    |
| CTRL | 0.72   | 0.51 |
| thkDA| 0.62   | 0.4  |

**Central Arctic**

|      | Extent | IIEE |
|------|--------|------|
| OBS  | 2.93   | -    |
| CTRL | 1.38   | 1.58 |
| thkDA| 2.33   | 0.7  |

**Atlantic sector**

|      | Extent | IIEE |
|------|--------|------|
| OBS  | 1.19   | -    |
| CTRL | 0.22   | 1.02 |
| thkDA| 0.83   | 0.53 |

*Figure 1: September-mean Arctic sea ice extent for the CMEMS reanalysis (using OSI-SAF) compared with modelled extent and ice edge error (IIEE) from the control (CTRL) and thickness assimilating (thkDA) seasonal predictions. Data are shown for 3 regions – distinguished by the underlying shading and corresponding box colours – as follows: combined Beaufort + Chukchi Seas (yellow), combined Kara, Laptev and East Siberian Seas (dark blue) and the central Arctic (red). Areas are defined using the standard NSIDC regions and units are millions of square km. Also shown (pink boxes) are corresponding statistics for the Atlantic and Pacific sectors of the Arctic Ocean. These are defined by splitting the union of the three coloured regions (i.e., red + yellow + blue) in two along the 30W and 140E longitude lines (yellow line) – which roughly follows the Lomonosov Ridge.*

---

## Author Comment (AC2) · 3 Aug 2018

**Response to review #1**

We thank the referee for their review of our manuscript. Answers to the specific comments in the review are listed inline below. The review comments are in black italics and our response is in red.

While we have taken the reviewers comments seriously, it is clear that many of the comments stem from a misinterpretation of the goal of this paper. Our goal here is to show the impact, on coupled seasonal predictions, of including CPOM CS2 sea ice thickness observations within the initialisation of the GloSea coupled model. Whilst the production of a sea ice analyses, for use as initial conditions, was therefore crucial to this stated goal, it was not, unlike for Allard et al. (2018), the primary purpose of this manuscript.

Having shown the usefulness of the sea ice thickness initialisation on the seasonal forecast system, we have laid the groundwork and motivation for a proper data assimilation treatment to use sea ice thickness observations in an ocean and sea ice analysis. That work would require the sort of data comparison with which many of the reviewer's comments are concerned. However, an in-depth evaluation of the thickness analyses, and comparison, of several sea ice thickness observational estimates, is outside the scope of the current manuscript.

We shall make these points clearer in the revised manuscript to ensure that the motivation for this study is clear.

*A general gripe of this study and similar ones: I find it really challenging to understand what you mean by the use of forecast and hindcast throughout the paper and they seem to be used interchangeably. If you are using prescribed atmospheric forcings that have assimilated real data then my view is anything using that is a hindcast not a forecast. Prediction is the more general term that could be appropriate but I see nothing in this paper that resembles a true forecast (no future knowledge), despite the title.*

We would disagree about the definition of 'hindcast' here. A hindcast is simply a retrospective forecast - performed under the same conditions as a true forecast but done when the result is already known. For example, when we perform hindcasts to test the forecast skill of the FOAM ocean-ice only forecasting system (Blockley et al., 2014), this is done using atmospheric forcing that has not assimilated data. However, this distinction is not relevant here because we do not use any atmospheric forcing for the coupled atmosphere-ocean-sea ice-land seasonal hindcasts performed within this study. What is important here, is that a hindcast is used to test the expected skill of a real forecast – and must be done in a fashion that does not use **any further** prior observational data after initialisation so as to invalidate that expectation.

As well as defining whether the prediction is made for a known past state (hindcast) or an unknown future state (forecast), the terms hindcast and forecast are also used within the GloSea seasonal prediction system (see MacLachlan et al. 2014) as a technical distinction. GloSea forecasts are carried out each day (2 members per day for the 210-day forecasts), whereas hindcasts – used for testing purposes and for the bias correction – are carried out for only 4 start dates per month but with more (8) ensemble members. This technical distinction has not been explained in this manuscript (although it is in the cited material) because it is not relevant to understanding out results.

We acknowledge that the terms 'forecast' and 'hindcast' have been used somewhat interchangeably in this manuscript, and that this may cause confusion. The reason for this is

that 'forecast' is much more easily understandable than 'hindcast' or 'retrospective forecast' - even though technically all coupled predictions in this work are the latter.

To make the forecast/hindcast distinction clearer we shall change our terminology to use 'predictions' throughout this study (including in the title). However, the use of `forecast' as an adjective (e.g. forecast skill) will remain. We hope this will help to make the distinction that these are unforced free-running seasonal coupled predictions.

**Main comments**

*It's really not clear from the motivation what it is you are trying to achieve by assimilating sea ice thickness in this study. In some cases you say the ice area/extent impact is negligible (as you ignore lower thickness ice) but other times it seems you highlight big improvements in your ice edge 'forecast'. You should really present a hypothesis you are testing in this kind of study (i.e. which metrics you are assessing). In general I would think assimilating thin ice should be especially important for seasonal (spring/summer) forecasting as you want to correctly incorporate this into your model as this is the ice most likely to melt out. If you get that wrong, you get the summer melt wrong. It seems like you used the opposite reasoning to justify not using the AWI/SMOS data but your results suggest the opposite to be true if you care about both the summer ice thickness and ice edge.*

We find it hard to follow the reviewer's line of argument here. We had hoped that the motivation for the study would be clear from the manuscript title: We show improvements in seasonal predictions of Arctic sea ice that arise from incorporating observations of sea ice thickness in the initialisation of these predictions. One of the more relevant predictions of ice cover is the ice extent, and ice edge location, at the end of the season (summer) – which is very much centre point in this manuscript – and shows significant improvement. Motivation for using these methods of success are that these quantities are easily assessed using current satellite concentration data, and that September-mean extent is the primary focus of the SIPN Sea Ice Outlook (although regional forecasts are also now gaining a focus).

As suggested, we shall add a hypothesis statement at the end of Section 1 to make our motivation for this study clearer and will review the abstract in this regard.

*I had a lot of issues with Section 2.2 (describing the CryoSat-2 data):*

*- You need to make clearer the various thickness datasets available and how a number of groups are now routinely producing sea ice thickness estimates, e.g. NASA/AWI/CPOM for CryoSat-2. - CPOM and NASA data were used in the study of Allard 2018. You need to make this clearer. It was in general unclear how your study differed from Allard and I think you need more discussion of their approach and results.*

Allard et al, (2018), further backed up by Stroeve et al, (2018), show the differences between the various datasets are not particularly major. Differences in the CS2 products arising from the retracking and processing algorithm differences are very small in comparison to the differences that nudging to CPOM CS2 data has on our sea ice thickness initial conditions. We shall add wording to this effect in our revised manuscript.

As our focus here is on the large-scale impact of initialising thick Arctic sea ice from CS2 within a fully coupled seasonal prediction system (GloSea). This is very different from the work of Allard et al. whose focus is on a reanalysis of sea ice thickness made using a forced oceansea ice model. An in-depth evaluation and comparison of sea ice thickness observational estimates, is outside the scope of our manuscript but is comprehensively addressed in Rick's. Having no desire to repeat the work already done in the Allard et al. (& now Stroeve et al.) studies, we have **chosen** here to test/show the impact of assimilating CPOM CS2 data in a seasonal prediction system.

The results of this study motivate us, and presumably others, to assimilate SIT properly within our operational sea ice analysis and seasonal prediction systems by making changes to the data assimilation system. During this next stage of the development process, we will absolutely care about things like observational error characteristics, the impact of choices made in the processing algorithms, and other details of the observation processing. This work is being addressed as part of our contribution to the H2020 SEDNA project. However, at this stage, for this feasibility study, we feel it is not relevant.

A statement of our proposed motivation and/or hypothesis will be added at the end of Section 1 to make these points clearer. We shall also include a sentence to state that these aspects were dealt with by Allard et al. (and add Stroeve et al.) and will also highlight how our work here differs from the study of Allard et al. (2018).

*- Is the pole hole and the data uncertainty really why these data haven't been used? It seems like any reasonable assimilation scheme shouldn't need complete coverage and can factor in data uncertainties. My guess is that the main reason was data availability, the fact these thickness products were in their infancy, and inertia.*

Yes data availability is also important but so is the pole-hole. Prior to CS2 the pole hole was very large. For ERS 1&2 it essentially covered all the central Arctic basin. A large pole-hole is not a big problem for the SIC assimilation because, historically at least, the ice is, fairly uniformly, close to 100% concentration near the North Pole. However for SIT, the situation is not so simple because thickness gradients across this region are quite large, meaning that any DA scheme attempting to spread the information would be heavily reliant on models to get this right. Coupled with the observational uncertainty issues this makes for a daunting problem.
However, we agree that availability is likely the most important factor here so we shall switch the order of the 3-fold reasons to put data availability first.

*- On that note, how do you treat the fact you are unlikely to have complete coverage from the CryoSat-2 data?*

We only modify the thickness fields where we have data and no attempt is made to spread observational data. This is fine for our feasibility study with monthly binned data, but when we move on to assimilating the raw altimeter tracks in a full variational scheme, much more work will be required to specify observational and model errors, covariances, etc.

More information about what is required for a full operational implementation of SIT assimilation will be added to the revised manuscript – briefly in the motivation and then again in the 'further work' section.

*- After listing problems associated with generic thickness data you then say this is improved by the availability of CS-2 thickness data. This doesn't make much sense as it is written.*

The availability of CS2 data does reduce the magnitude in all 3 of the problem areas we highlight (timeliness, pole-hole, accuracy). Most apparent is the reduction in pole-hole diameter with CS2 owing to the orbit inclination. However, the accuracy is also improved owing to the higher along-track accuracy/resolution of the SIRAL altimeter (compared with ENVISAT & ERS-1/2). (See for example Guerreiro et al., (2017) who use CryoSat-2 to correct biases in Envisat freeboard estimates).Finally, as the SIT methods have become more mature, their processing has become quicker and more operationally robust. We now have access to CS2 SIT from CPOM in near-real-time which make SIT a viable option for incorporation into our operational prediction systems.

This is partially explained in the following sentences but we shall modify this text to explain these points more clearly and add the Guerreiro et al., (2017) reference.

 *- You should cite the relevant studies regarding uncertainty estimates, not just the Ricker/AWI reference and apply that to CS-2 data derived by other groups. There are strong differences in the retracking procedures which may have impacts on respective data uncertainties across the products.*

As we have previously stated, we are not concerned about the differences between these observational products in this study. Instead our motivation is the impact of SIT initialisation on seasonal predictions of Arctic sea ice cover (i.e., extent and ice-edge location). We use the – really quite comprehensive – studies of Ricker et al. here purely to provide approximate bounds below which the CS2 is likely to be of no value. However when we come to do this as a data assimilation problem – rather than a seasonal coupled prediction problem – we will be very interested in the observational uncertainty/properties.

 *- I don't get what this extra quality checking of the CPOM CS-2 data is. The fact you have included a personal comms from one of the data producers of that dataset makes it seem like this is something they do too? What exactly do you mean by smearing?*

This is related to the regridding and binning of the data performed by CPOM to ensure that high spatial gradients are not smeared out. This was actually recommended by CPOM and so should be considered part of their observational processing.

With this in mind, we now feel that the inclusion of this sentence is actually distracting to the reader – especially given that observational uncertainty is not our primary concern in this study. We therefore propose to remove this sentence from the revised manuscript.

*Why not use daily along-track CryoSat-2 data? I thought this was the whole purpose of CPOM releasing the daily along-track data? Instead they grid the data, then you grid the data, then you interpolate to a get a daily thickness?*

Yes, you are correct. Assimilating altimeter tracks of thickness (or more likely the raw freeboard) is the ultimate goal for SIT assimilation in our systems. However much work is required to do this. Observational errors need to be quantified (including representativeness), and model/observation covariances and correlations are required to spread the data from the tracks. Finally, balancing with ice concentration and other aspects of the model (SST, SSS) are required. Doing all this is a considerable undertaking and so, before doing this, we wanted

to be sure that the SIT initialisation would have an impact on the model – hence this feasibility study using gridded data.

We shall make this point clearer in the introductory motivation – namely the hypothesis statement at the end of Section 1. More information about what is required for a full operational implementation of SIT assimilation will also be added to the 'further work' section of the revised manuscript.

*Hard to tell what this volume comparison really means. You compare with PIOMAS but then say that data is biased low (which I'm not actually sure is true when you look at more than one CS-2 estimate) so it is actually good that you are further from that data? You say this was expected but this seems like a hindsight statement to me. I agree PIOMAS data can provide useful context but I don't agree with how you've used it. I think you should just show the CS-2 data and say look, the assimilation does what it is supposed to do.*

Yes, it was expected that the winter sea ice volume would be higher than PIOMAS. This was an obvious and logical expectation given that the CPOM CS2 data we are assimilating has higher volume than PIOMAS (as documented by the Laxon et al. and Tilling et al. CPOM studies). However, the key point here is that this doesn't actually matter. We only include PIOMAS comparison in our evaluation as a reference because it is well understood and widely used for this purpose. We do not use it for verification. We shall make this clearer in the revised manuscript.

Furthermore, we shall take your final piece of advice to illustrate the impact of the assimilation by showing that the CS2 analysis matches the CS2 data. This will be done by extending Figure 2 to include the CS2 observations and the model fields (rather than just the differences).

*P10 L10-20 and elsewhere: Very confusing to me if these are hindcasts or genuine forecasts. You use both labels interchangeably. How could you move forward to produce genuine forecasts?*

As stated earlier, these are hindcast or retrospective forecasts that are identical to genuine forecasts in everything except the fact that they are performed in the past (i.e., with the result already known). A 'hindcast' is a prediction made for a known past state whereas a 'forecast' is made now for an unknown future state. All other aspects of the prediction are essentially identical. To produce genuine forecasts we would just need initial conditions for now and then wait 4 months to see how well they did.

We shall modify the manuscript to say 'prediction' instead of either forecast or hindcast, which were used somewhat interchangeably.

**Specific comments**

*In the abstract: really the 'first time, we directly assess the impact of winter sea ice thickness initialisation on the skill of seasonal summer forecasts'? Do you mean in the Met Office model framework? I think Allard and others have done this and I also don't think you do assess forecast skill*

Yes, this is the first documented study to assess the impact of using satellite sea ice thickness data to initialise a fully coupled seasonal prediction system. This fact is confirmed by the

comments of referee #2 who states, "This study represents the first known fully coupled atmosphere-ocean-ice forecast system to utilize CryoSat-2 ice thickness data for seasonal forecasts".

The study of Allard et al (2018) is very different because they use a forced ocean-sea ice model to perform long ocean/sea ice analyses. They also perform a thorough assessment of the thickness analyses produced and consider short-range, uncoupled, forecasts. Here we are only interested in the analysis in the context of providing sea ice initial conditions for our seasonal predictions – made using the GloSea seasonal prediction system.

We are a little confused about the referee's comment "I also don't think you do assess forecast skill". Here we run 24 retrospective seasonal forecasts per year, for each of 5 years (120 in total), to produce seasonal predictions of September Arctic sea ice using initial conditions in April/May. These predictions are performed under forecast conditions using an initialised coupled climate model in the same way as we do operationally in the GloSea Seasonal Prediction System. We then evaluate the quality of September-mean Arctic sea ice predictions by comparing basin-wide extent against observational estimates (from NSIDC, HadISST & OSI-SAF – the latter using the CMEMS reanalysis), and by calculating integrated ice edge error (IIEE) against the latter dataset. This is the standard methodology for examining skill in a seasonal forecast system, where the accumulation of sufficient evidence from "real time" forecasts would require substantial delays to the provision of information required to update forecast systems within a realistic period.

*Introduction - In general you need more updated references. A lot of this discussion is a bit outdated now. i.e.:*
 *- Drop the Vaughan/IPCC refs and use the more specific refs. Try Serezze & Stroeve 2015 for a more recent seasonal sea ice trends citation?*
*- The Collins/IPCC is also a bit outdated. I think you can add some of the more recent references to sea ice projections - e.g. Jahn 2016, 2018.*

We believe the most appropriate citations are those given. The intention here is to motivate the fact that Arctic sea ice has declined/is in decline and that is projected to continue.

The most robust evidence of this is provided by these multi-author IPCC references, which are created by a multi-disciplinary (multi-centre, multi-country, etc.) team of authors. (This is also true for the very comprehensive, multi-author study of Meier et al. (2014).)

*P2 L1-11 - I think this is not useful information as it is not that relevant to seasonal forecasting and a lot of the references and discussion are outdated. Either update/improve or drop.*

We disagree and think this is extremely relevant material. Substantial resources are being invested in seasonal planning in the Arctic (e.g. within the EU's Horizon 2020 programme, the ARCUS Sea Ice for Walrus Outlook (SIWO), in support of projects endorsed by the WMO's Year Of Polar Prediction (YOPP) & MOSAiC), with more such investment by both government and private organisations is likely in the future. In particular, there are a number of projects endorsed by YOPP which focus on sea ice seasonal prediction (you can see an overview of YOPP-endorsed projects at https://apps3.awi.de/YPP/endorsed/projects, noting in particular: https://apps3.awi.de/YPP/pdf/stream/79, https://apps3.awi.de/YPP/pdf/stream/100, https://apps3.awi.de/YPP/pdf/stream/106, and https://apps3.awi.de/YPP/pdf/stream/172).

We do not believe that, as a general rule, citations should have a "best before date". So if no further information has been published in the meantime to contradict the findings of these papers, which clearly have had a large impact on the funding agencies, then they are entirely appropriate.

*P2 L13 - change sentence ordering.*

OK. We shall change this to: "Interest in seasonal predictions has increased following the drastic reduction in Arctic sea ice extent in the summer of 2007, which led to a (then) record-low summer minimum extent being set."

*P2 L17 - the predictive skill sentence is confusing. SIPN haven't really assessed that.*

Yes it is true that SPIN, the US project, has not done this itself. However, our point here is that the existence of SIPN has caused this as a secondary effect. The community that has been built up around SIPN has fostered collaborative studies in this area, which have been supported by the forecast data provided to SIO (and the models used to produce them). We shall reword this to make the distinction clearer.

*P2 L19-23 - I don't think you've really said why it is interesting though! Either make a clearer point regarding its scientific interest (e.g. what the predictability/memory of the system is compared to other components of the climate system).*

Well it interests us and, given the number of papers published on the subject, we suspect many others too. We shall reword this sentence to emphasise the fact that prediction beyond medium-range timescales is challenging.

*Also I don't think it is clear that sea ice is now necessarily harder to predict. Having some enhanced variability may be useful. Your figure 4 doesn't show an increase in ice edge error for instance!*

In fact Figure 4 does have an increasing trend!  The IIEE in the control predictions (Fig 4b) is increasing over the period 1992-2015 with a small slope of approx. 0.0087 million square km per year (or 8700 square km per year). Although this is small relative to the long-term mean, it is statistically significant (p-value < 0.016). We include a modified version of Fig 4b below with the trend line overlain in red (see Figure 1 below). Of course, this in itself does not prove that forecasting is more challenging with less/thinner/more variable sea ice – and we are not arguing here that it does. However thinner sea ice will be more heavily influenced by the non-linear, chaotic atmospheric circulation (both dynamically and thermodynamically), which would undoubtedly be less predictable.

That prediction of sea ice becomes harder as the ice thins is one of the results of Holland et al (2010) who show that "ice area in a thicker sea ice regime generally exhibits higher potential predictability for a longer period of time". Furthermore Stroeve et al (2014) also support this stating: "The reduced predictive skill as the winter ice cover thins has been noted in some of the contributions to the SIO and appears to be coincident with the rapid thinning of the ice cover."

We shall add additional citations to Holland et al. (2010) and Stroeve et al. (2014) at the end of this sentence.

*P2 L27 - why exactly does a lack of observations make the forecasts harder? Less to assimilate in models or to validate? Below you list a number of observations that are available in the poles...*

We did not mean to imply that the forecast models were hampered by the lack of observations, but that it is the initialisation of forecasts that are hampered by the lack of observations. We shall change this wording to: "In particular, initialisation of forecasts in the Arctic are less accurate owing to observations being less abundant, and assimilation techniques less advanced in polar regions, hampering the forecasts in these regions as compared to forecasts at lower latitudes."

*P2 L35 - include acronym definitions.*

Yes, this is the first usage of the acronym FOAM = "Forecast Ocean Assimilation Model" in the paper. This definition shall be moved forward from later in the paper (p. 4, l33).

*P3 L4 onwards - this is a bit of a confusing paragraph to me. What is the point you are trying to make? In general my view is that there is hope for dynamical models being used for skillful sea ice forecasting based on some of the perfect model studies that you cited. However the SIO has really shown that they are not currently performing much better than the linear trend in many cases (as shown by Stroeve et al., 2014).*

Yes this is pretty much the point we are trying to make. We shall reword the motivation in Section 1 to make this clearer.

*It still seems that dynamical models are lagging behind more simple statistical methods (e.g. Schroeder et al., 2014, Petty et al., 2017)*

Yes this may be true. The studies of Stroeve et al. (2014) and Hamilton and Stroeve (2016) suggest that predictions from both statistical and dynamical models beat heuristic predictions (guesses) and that this is statistically significant. They further suggest that statistical models are slightly better than dynamic but not with any degree of significance. One could therefore conclude that it seems fairly simple to create a statistical model that can predict Arctic sea ice extent to a similar degree as dynamic models. However, although statistical models are interesting (and promising), they do not replace the need for the full dynamical models which provide a much wider offering. For example creating a statistical model that would predict the ice edge 5 months in the future, with anything like the degree of accuracy seen in the predictions we make here, would be a very challenging endeavour.

*I think you need to add in some comments on the different forecast methods available, merge with the following paragraph about improving dynamical models and make clearer what the motivation of this study is! This should be the key paragraph of the introduction.*

*P3 l 24 - not sure how this point links to the above.*

We shall modify the introductory motivation in Section 1 to make this (and the 2 previous point) clearer and to better motivate the interest in seasonal predictions using dynamical models.

*P4 L1-4 - but in the abstract you imply you are the first to do this?! I guess you meant in your fully coupled Met Office forecast framework. You need to make that clearer.*

Yes we are. As stated previously this is the first use of satellite thickness data to initialise seasonal coupled predictions of Arctic sea ice. The previous studies listed here are using forced ocean-sea ice models and performing reanalyses and/or short-range forecasts.

In contrast we are performing seasonal predictions (5-month forecasts) using a fully coupled atmosphere-ocean-sea ice-land model with the GloSea seasonal prediction system.

*P5 L11-13 - why just mention the ocean reanalysis component here? Would be more understandable if you referred to GloSea as a reanalysis.*

No! GloSea is a coupled seasonal prediction system (see MacLachlan et al (2014)). Referring to it as a reanalysis would be more misleading.

The motivation in this section of the manuscript is to introduce the fact that long reanalyses are performed using an offline analogue of the FOAM ocean analysis system. These long reanalyses are used within the GloSea seasonal prediction system to initialise hindcast (or re-forecast) experiments. They are also used for furthering understanding of the ocean and how the ocean has changed over the satellite period (e.g. within ORA-IP). As the ocean reanalysis is used within the GloSea seasonal prediction system, it is sometimes (erroneously/unfortunately) referred to as the GloSea ocean re-analysis. However, GloSea is a coupled global seasonal prediction system (see MacLachlan et al., 2014).

We shall reword these parts of Section 2 to make this clearer to the reader.

*P5 L23 - I don't think these are the correct citations here. Link to relevant passive microwave concentration datasets instead or recent papers describing that long-term record (e.g. Parkinson/Comiso papers).*

The motivation behind these citations is two-fold: 1) to provide references for the HadISST and NSIDC data sources that we use within our forecast evaluation; 2) to show that we have long-term sea ice concentration observations from multiple sources. The citations given here are those recommended/requested by the data providers for the NSIDC sea ice index (https://nsidc.org/data/g02135) and for the HadISST dataset (https://www.metoffice.gov.uk/hadobs/hadisst/). Therefore, their inclusion is required for our objective #1 and we re-use them for our objective #2 to avoid including lots of similar citations.

*P5 L25 - should reference Kwok and Cunningham 2008 instead.*

We shall change this from the 2009 reference to the Kwok and Cunningham (2008) reference.

*P6 L16-17 - what do you mean by sensitive here? I think you mean uncertain/challenging? Again, is there nothing in one of the CPOM papers that highlights this issue?*

Yes we mean uncertain. Sensitive will be replaced with uncertain in the revised manuscript.

When freeboard is very low it is difficult to distinguish from SSH fluctuations and gravity waves. This issue is well documented in the comprehensive Ricker et al. citations that we have used. Given that the issue is related to the whole process of deriving freeboard from satellite altimetry – in particular CS2 SIRAL radar altimeter which penetrates the snow and into the upper layer of the ice – rather than the centre who happen to be processing the data, we are happy that these citations cover the issue adequately.

*P6 L30 - I think this is a big guess. Do we really know much ice < 2 m melts away each summer? That Keen modeling study (Fig 2?) suggests some 40% of the ice less than 2 m (including ice and snow in that thickness) does not melt through in summer. A lot will have to do with how much snow there was on the ice and where/when melt onset occurs. Even if it does melt away, this seems to be crucial information for determining solar absorption that can drive SST increases and further sea ice melt.*

This paragraph has been removed in revision. The hypothesis/motivation statement at the end of Section 1 will tell the reader that our interest is the impact of thick ice initialisation on seasonal prediction skill. To avoid any confusion with thin ice we shall no longer refer to SMOS in this section.

Instead, we shall start subsection 2.2.1 with some additional motivation for our thick ice initialisation along the lines of: "For accurate seasonal predictions of September sea ice cover it is important to model ice that will persist throughout the summer season. Meaning that an improved representation of the location of thick sea ice within the initialisation of our system should be advantageous. In this study we shall initialise our model using thick ice from CS2, which are accurate for ice thicker than 1m (Ricker et al., 2017). We shall use monthly CS2 winter (Oct-Apr) thickness estimates produced by CPOM (Tilling et al., 2016) which start from October 2010 until present (at time of writing)…."

*Also need to make the point here (and earlier) that AWI do produce a merged product!*

We find this statement hard to reconcile with the referee's earlier comments on using a gridded ice thickness product as opposed to the along track product.

However we shall again emphasise here that we are using the CPOM CS2 product as a test bed to investigate the effect that initialisation of sea ice thickness will have on a coupled seasonal forecast. We are not performing a summary, or comparison, of the available thickness data products.

Our results here motivate the desire to implement sea ice thickness observations properly within our 3D-Var assimilation framework. As part of this extension, we will be combining observations of sea ice thickness derived from both altimeter (CS2) and radiometer brightness temperature (SMOS) measurements, in a synergistic fashion, within the data assimilation system.

Because this is a blended product (created using an optimal interpolation assimilation technique), it is not something that we would normally assimilate in the system. We do however use gridded analyses like this for the purposes of model evaluation and will likely use this CS2SMOS data in the next stages of this work.

*Section 2.3 - Why do you need extent and concentration? Surely you are just assimilating sea ice concentration? This needs to be made much clearer here.*

*There appears to be some confusion by the reviewer that this section is referring only to data used for assimilation. Section 2 is about all the models and datasets used in this study – both for assimilation and for evaluation. Although Section 2.3 covers the validation datasets in paragraph 1 and the assimilation data in paragraph 2, we acknowledge that it is not very well explained.*

*Therefore we shall alleviate this confusion in the revised manuscript by explicitly stating this in the preamble to Section 2.3. We shall further modify paragraphs #1 and #2 to state that data is used for validation and assimilation respectively.*

*- The use of CMEMS 'data' seems very confusing to me. Why not use observed ice concentration?!*

It is very common to use an analysis product for evaluation purposes. For example, many people use the OSTIA analysis for evaluating SST or ERA-Interim for evaluating atmospheric variables. Likewise, people are now starting to use the blended CS2SMOS product (Ricker et al. 2017) to evaluate sea ice thickness (which itself is an SIT analysis created using O/I assimilation methods). Although we would never assimilate analysis products like these, we do use them for the purpose of model evaluation.

In this study we validate using the ice concentration from the CMEMS reanalysis, which is an analysis made using the OSI-SAF sea ice concentration data. The main reason for using this is that it is already on the correct model grid. We state that: "Using this CMEMS reanalysis has the benefit that it is performed on the same ORCA025 grid as the ocean-sea ice components of the GloSea seasonal forecasting system, which makes spatial comparisons easier."

It is important to note that, except for the differences due to the spatial resolution, the sea ice concentrations in the OSI-SAF observed data and the CMEMS analysis are virtually identical. The CMEMS product can therefore be thought of as a dynamically consistent re-gridding. It is of course very important to compare extent using products on the same grid because extent, as a metric, is very much dependent on grid/resolution (Notz, 2014).

*- The NSIDC sea ice index is just a monthly index of total ice extent. This isn't what you use, right?*

Yes we use the NSIDC single number extent for the purposes of validating our September-mean ice extent predictions (in Fig 4). However we also use extent derived from HadISST sea ice concentration, and from the CMEMS reanalysis (i,e., OSI-SAF).

*P8 L6 - what is the size of your model grid (in kilometers?) how does this translate to the 5 km CPOM data?*

The ORCA025 tripolar grid was created to avoid the singularity associated with the convergence of meridians at the North Pole, which it achieves by defining two distinct north

poles over Canada & Siberia. The ocean points in between form a variable resolution grid with highest resolution nearest the two poles i.e., in the Canadian Arctic Archipelago and the Laptev Sea. The resolution in the Arctic Ocean ranges from ~9km up to ~15km.

*P8 L14-24 - Think you should list out the CICE thickness categories. How else could you have done this? I think it would be worth presenting more sophisticate approaches for future work, however I get that you started with this simple approach.*

We use standard WMO categories that are one of default options within the CICE model. These are listed in the supporting Blockley et al., (2014), and Ridley et al. (2018) references.

In the revised manuscript we shall include this information in Section 2.1 where referee #2 has requested some additional model information.

More sophisticated approaches will be covered in the 'future direction' paragraph in Section 5 where, in line with other comments above, we shall be including more information about what is required to implement SIT assimilation operationally.

*Figure 2 - I think you should also show (maybe in the supplementary info?) what the pre nudged, and nudged thickness fields are. Could just do this for the mean October-April thickness and also update Figure 1 to show this longer season too. If the mean thickness was way off before it makes sense that assimilating the thickness will improve things.*

We are not entirely sure what the reviewer means by pre-nudged and nudged thickness fields. Because the nudging is so small and applied every time-step pre-nudged and nudged fields will look virtually identical.

The proposed changes to Figure 2 should better show the impact of the nudging as it will include plots of mean thickness for the control and thkDA that can be compared with the CS2 observations.

*Why the different start dates for the forecasts? Pretty confusing.*

This is how the GloSea seasonal prediction system works. GloSea runs every day at the Met Office and produces 2 forecasts of length 210-days. These are used with forecasts from previous days to create a large, lagged ensemble of forecasts. For the hindcasts/reforecasts performed here, an 8-member ensemble is run for fewer distinct start dates. This is done to be consistent with the way that GloSea performs its operational hindcasts – which are initialised from 4 start dates per month (see MacLachlan et al).

*Figure 4 is hard to see. Maybe box plots of the recent years showing the variability in the different estimates in the different years?*

Referee #2 also asked for the layout of Figure 4 to be modified. We are currently trialling options and the revised manuscript will have an improved Figure 4 that will be easier to read. More space on the plot will be devoted to the key time-period of 2011-2015, and we will ensure that the individual ensemble member values are distinct (either by using a box and whisker approach as the referee suggests, or by better spacing out the existing crosses).

*P11 L10 - CMEMS isn't really an observational estimate, right? Based on the assimilation of OSI-SAF...*

We shall change this sentence to refer to CMEMS as a reanalysis product: "To assess the accuracy of the GloSea seasonal predictions, observational, and reanalysis, estimates of Arctic extent, from the CMEMS, HadISST and NSIDC sources (see Section 2.3), are plotted alongside the model predictions (black/grey)."

We note again that, up to issues due to resolution differences, the sea ice extents estimated directly from OSI-SAF and from the CMEMS reanalysis are virtually identical. The CMEMS analysis being on the same grid as the forecast ice concentrations presented in this study makes it the most comparable analogue.

*Pll L14 - not sure what you mean by 'building a picture' here. I see no value in showing that earlier data.*

We agree that the picture requires more building and so we shall modify the text to better paint this picture.

The important features here are that the long control set of re-forecasts exhibits several features: the IIEE is virtually flat (although slightly increasing); the extent is consistently biased low. This shows us that the model – without SIT assimilation– is consistently wrong over this long period, which is more powerful/useful than just looking at a short 5-year section.

*P11 L17 what is close*

We shall update the manuscript to include the differences (i.e., like "2011 is X rather than Y and 2012 is X2 rather than Y2".

*Figure 5 - include numbers on map. Plot the IEEE as a time series.*

We shall reprocess Figure 5 so each plot includes a label along the lines of "Extent (model) = X; Extent (obs) = Y; IIEE = Z".

However, we are somewhat confused about the latter part of the referee's comments here because Figure 4b already shows the time series of IIEE.

*Figure 7 and 8 - I don't understand these maps. How exactly is the data shown in Figure 8 calculated? Also is Figure 8 averaged over the entire year, but figure 7 is September? Why are the pressure units different? Perhaps better to show Figure 7a and 8a together, then 8a and 8b*
*The discussion of using ensemble members instead of the ensemble means and how this relates to assessing model bias was confusing and needs more description/clarification.*

Apologies. It looks like "September" was accidentally left off the Figure 8 caption – which we shall fix. We also agree that switching these figures over makes the arguments easier to comprehend. So in the revised manuscript we will have: Figure 7 showing T2M (differences and RMSE), and Figure 8 showing the same for pressure. We shall also extend Fig 8 to include both MSLP and Z500 to prevent confusion between the two.

We shall also update the text in Section 4.2 to better explain how RMSE etc. are calculated.

*Should either enhance the CLIM analysis or drop. i.e. repeat with using a fixed (think FIXED_IC would be a better acronym) for all years of data available.*

We agree that "FIXED-IC" is a better label. In the revised manuscript we shall change "CLIM-2015" to "FIXED-IC".

*How do these results compare with Allard?*

Our study is very different from Allard et al. As stated previously, we are initialising a fully-coupled atmosphere-ocean-sea ice-land (AOIL) model with CryoSat-2 (CS2) sea ice thickness and using it to perform an ensemble of seasonal predictions from May through to September. We find that initialising with CryoSat-2 SIT gives us a considerable improvement in Arctic sea ice extent and ice edge location. We also show that memory of winter thickness changes in the initialisation carry through to the end of summer.

Meanwhile Allard et al. (2018) performed an 18-month reanalysis using an ocean-sea ice model forced by atmospheric analyses. They assimilate CS2 thickness during the winter when the data is available. In contrast to our study, and by virtue of the fact they run a reanalysis, they continue assimilating all other variables throughout the year. They show a considerable improvement in their winter thickness analyses when using CS2.

What the two studies have in common is that:

1) we both show that assimilating sea ice thickness provides an improvement (us: in coupled seasonal forecasts; them: in their forced reanalysis);
2) we both show that memory of initialised winter thickness is still present in the summer (us: after a 5-month free-running coupled forecast; them: after 5 months of assimilating everything except SIT in a forced reanalysis).

Both studies show that sea ice thickness initialisation is beneficial – albeit in very different setups.

*P14 L21 - not sure what you mean by work well. Maybe not gone wrong?*

Technically, the application of thickness increments within the CICE sea ice model while performing a sea ice analysis has worked exactly as expected. The increments have been retained by the model and produce thickness initial conditions, for initialising the seasonal forecasts, that are much closer to the CS2 data. There is also the matter of the exceptional improvement to sea ice location (IIEE) when initialising with SIT.

These points will be better described in the revised manuscript. The fact that the analyses are much closer to the CS2 observations will be illustrated by the proposed changes to Figure 2.

*P14 L1 Why is this from dynamics not thermodynamics?*

The distribution of thickness across the Arctic is caused by a series of dynamic processes. The ice motion, primarily driven by the winds, consists of a re-circulation around the Beaufort

Gyre with a transpolar drift which drives across from Siberia to the north coast of Greenland and north of Svalbard. This causes thick ice to pile up north of Greenland. The fact that we have thicker ice dragged around into the Beaufort Sea and not enough thin ice north of Fram Strait suggests that the ice is too mobile. Both the proposed EAP rheology and form-drag changes would reduce the ice speed and should reduce the bias.

This is also a well-known bias in the sea ice modelling world and is present in most models – including most CICE models, NAOSIM, PIOMAS etc. (see for example Lindsay et al. (2012)).

There is of course a small chance that this is thermodynamically driven, which is why we say this is "most likely" caused by deficiencies in the dynamics.

*Also drop 'so-called ice-ice force'*

This will be dropped in the revised manuscript.

[Figure]

*Figure: The original Figure 4b with the IIEE trend line overlain in red.*

**References**:

Guerreiro, K., Fleury, S., Zakharova, E., Kouraev, A., Rémy, F., and Maisongrande, P.: Comparison of CryoSat-2 and ENVISAT radar freeboard over Arctic sea ice: toward an improved Envisat freeboard retrieval, The Cryosphere, 11, 2059-2073, https://doi.org/10.5194/tc-11-2059-2017, 2017.

Lindsay, R., Haas, C., Hendricks, S., Hunkeler, P., Kurtz, N., Paden, J., Panzer, B., Sonntag, J., Yungel, J., and Zhang, J.: Seasonal forecasts of Arctic sea ice initialized with

observations of ice thickness, Geophys. Res. Lett., 39, L21502, doi: 10.1029/2012GL053576, 2012.

Stroeve, J. C., Schroder, D., Tsamados, M., and Feltham, D.: Warm winter, thin ice?, The Cryosphere, 12, 1791-1809, https://doi.org/10.5194/tc-12-1791-2018, 2018.

---

## Author Response (AR1)

**Authors' final response**

We thank the referees for their review of our manuscript. Answers to the specific comments in the review are listed inline below. The review comments are in *black italics*, our response is in red, and details of manuscript changes are in blue.

Page and line numbers given here refer to the revised manuscript version with full mark-up.
* * *
**Response to review #1**

While we have taken these comments seriously, it is clear that many of the comments stem from a misinterpretation of the goal of this paper. Our goal here is to show the impact, on coupled seasonal predictions, of including CPOM CS2 sea ice thickness observations within the initialisation of the GloSea coupled model. Whilst the production of a sea ice analyses, for use as initial conditions, was therefore crucial to this stated goal, it was not, unlike for Allard et al. (2018), the focus of this manuscript.

Having shown the usefulness of the sea ice thickness initialisation on the seasonal forecast system, we have laid the groundwork and motivation for a proper data assimilation treatment to use sea ice thickness observations in an ocean and sea ice analysis. That work would require the sort of data comparison with which many of the reviewer's comments are concerned. However, an in-depth evaluation of the thickness analyses, and comparison, of several sea ice thickness observational estimates, is outside the scope of the current manuscript.

**Changes to the manuscript:** The 2 sentences in the penultimate paragraph of the Introduction have been expanded to form a new hypothesis paragraph. (P4 L17+) Additionally more information is provided in the Summary section about what would be required to implement SIT assimilation fully within the GloSea initialisation using the NEMOVAR 3D-Var assimilation scheme (P17 L11+)

*A general gripe of this study and similar ones: I find it really challenging to understand what you mean by the use of forecast and hindcast throughout the paper and they seem to be used interchangeably. If you are using prescribed atmospheric forcings that have assimilated real data then my view is anything using that is a hindcast not a forecast. Prediction is the more general term that could be appropriate but I see nothing in this paper that resembles a true forecast (no future knowledge), despite the title.*

We would disagree about the definition of 'hindcast' here. A hindcast is simply a retrospective forecast - performed under the same conditions as a true forecast but done when the result is already known. For example, when we perform hindcasts to test the forecast skill of the FOAM ocean-ice only forecasting system (Blockley et al., 2014), this is done using atmospheric forcing that has not assimilated data. However, this distinction is not relevant here because we do not use any atmospheric forcing for the coupled atmosphere-ocean-sea ice-land seasonal hindcasts performed within this study. What is important here, is that a hindcast is used to test the expected skill of a real forecast – and must be done in a fashion that does not use **any further**, subsequent, observational data after initialisation so as to invalidate that expectation.

As well as defining whether the prediction is made for a known past state (hindcast) or an unknown future state (forecast), the terms hindcast and forecast are also used within the

GloSea seasonal prediction system (see MacLachlan et al. 2014) as a technical distinction. GloSea forecasts are carried out each day (2 members per day for the 210-day forecasts), whereas hindcasts – used for testing purposes and for the bias correction – are carried out for only 4 start dates per month but with more (8) ensemble members. This technical distinction has not been explained in this manuscript (although it is in the cited material) because it is not relevant to understanding our results.

**Changes to the manuscript:** use of 'forecast' and 'hindcast' have been replaced with 'prediction' where appropriate throughout the manuscript (including the title). However, the use of 'forecast' as an adjective (e.g. 'forecast skill') has remained. We have also made it clearer that the GloSea seasonal predictions made here (and generally within the hindcast context) are done without any knowledge of future observations and so are indeed 'true forecasts'. (P5 L14; P11 L27-29)

*Main comments*

*It's really not clear from the motivation what it is you are trying to achieve by assimilating sea ice thickness in this study. In some cases you say the ice area/extent impact is negligible (as you ignore lower thickness ice) but other times it seems you highlight big improvements in your ice edge 'forecast'. You should really present a hypothesis you are testing in this kind of study (i.e. which metrics you are assessing). In general I would think assimilating thin ice should be especially important for seasonal (spring/summer) forecasting as you want to correctly incorporate this into your model as this is the ice most likely to melt out. If you get that wrong, you get the summer melt wrong. It seems like you used the opposite reasoning to justify not using the AWI/SMOS data but your results suggest the opposite to be true if you care about both the summer ice thickness and ice edge.*

**Changes to the manuscript:** As mentioned above, the 2 sentences in the penultimate paragraph of the Introduction have been expanded to form a new hypothesis paragraph. (P4 L17+) We have also dropped the additional motivation/justification for using CS2 rather than SMOS from Section 2.2.1 to avoid confusion. The story is now clearer: we are interested in the impact of thick ice initialisation on the skill of seasonal predictions of Arctic September sea ice cover and hypothesise that including CS2 within the initialisation will improve skill.

*I had a lot of issues with Section 2.2 (describing the CryoSat-2 data):*

*- You need to make clearer the various thickness datasets available and how a number of groups are now routinely producing sea ice thickness estimates, e.g. NASA/AWI/CPOM for CryoSat-2. - CPOM and NASA data were used in the study of Allard 2018. You need to make this clearer. It was in general unclear how your study differed from Allard and I think you need more discussion of their approach and results.*

Allard et al, (2018), further backed up by Stroeve et al, (2018), show the differences between the various datasets are not particularly major. Differences in the CS2 products arising from the retracking and processing algorithm differences are very small in comparison to the differences that nudging to CPOM CS2 data has on our sea ice thickness initial conditions.

As our focus here is on the large-scale impact of initialising thick Arctic sea ice from CS2 within a fully coupled seasonal prediction system (GloSea). This is very different from the work of Allard et al. whose focus is on a reanalysis of sea ice thickness made using an externally forced ocean-sea ice model. An in-depth evaluation and comparison of sea ice thickness

observational estimates, is outside the scope of our manuscript but is comprehensively addressed in Rick's. Having no desire to repeat the work already done in the Allard et al. (& now Stroeve et al.) studies, we have **chosen** here to test/show the impact of assimilating CPOM CS2 data in a seasonal prediction system.

**Changes to the manuscript:** We have expanded the introduction of Allard et al. (2018) to make it clear that they run an externally forced ocean-sea ice model and look at analyses and short-range forecasts. We further state that Allard et al. use multiple thickness datasets processed with different algorithms and perform an in-depth analysis of the data. We also explicitly state that what has not been previously investigated is the impact that assimilation of sea ice thickness may have in longer (>90 days), forecasts made using fully coupled systems. (P4 L7+) When introducing the CS2 data in Section 2.2.1, we have added a sentence stating that other centres provide CS2-derived Arctic sea ice thickness products. An additional citation to Allard et al. (2018) is included (who compare the different products) as well as a citation to Stroeve et al. (2018). (P7 L7-9) The new hypothesis/motivation paragraph in the Introduction, will tell the reader that our study is focussed on initialising thick sea ice using CS2 and testing the impact on seasonal predictions of September sea ice cover (and not a summary, or comparison, of available sea ice thickness datasets). (P4 L17+)

*- Is the pole hole and the data uncertainty really why these data haven't been used? It seems like any reasonable assimilation scheme shouldn't need complete coverage and can factor in data uncertainties. My guess is that the main reason was data availability, the fact these thickness products were in their infancy, and inertia.*

Yes data availability is also important but so is the pole-hole. Prior to CS2 the pole hole was very large. For ERS 1&2 it essentially covered all the central Arctic basin. A large pole-hole is not a big problem for the SIC assimilation because, historically at least, the ice is, fairly uniformly, close to 100% concentration near the North Pole. However for SIT, the situation is not so simple because thickness gradients across this region are quite large, meaning that any DA scheme attempting to spread the information would be heavily reliant on models to get this right. Coupled with the observational uncertainty issues this makes for a daunting problem.

**Changes to the manuscript**: The order that these 3 reasons are listed in has been switched to put data availability first.

*- On that note, how do you treat the fact you are unlikely to have complete coverage from the CryoSat-2 data?*

Using the nudging approach, we only modify the thickness fields where we have data (P9 L11-12) and no attempt is made to spread observational data. This is fine for our feasibility study with monthly binned data, but when we move on to assimilating the raw altimeter tracks in a full variational scheme, more work will be required to specify observational and model errors, covariances, etc.

**Changes to the manuscript**: More information is provided in the Summary section detailing what would be required to implement SIT assimilation fully within the GloSea initialisation using the NEMOVAR 3D-Var assimilation scheme (P17 L11+)

*- After listing problems associated with generic thickness data you then say this is improved by the availability of CS-2 thickness data. This doesn't make much sense as it is written.*

**Changes to the manuscript:** Section 2.2 has been modified to explicitly state how CS2 reduces the 3 problems we highlight (timeliness, pole-hole, accuracy). A new reference to Guerreiro et al. (2017) has been included. (P6 L25-28)

*- You should cite the relevant studies regarding uncertainty estimates, not just the Ricker/AWI reference and apply that to CS-2 data derived by other groups. There are strong differences in the retracking procedures which may have impacts on respective data uncertainties across the products.*

As we have previously stated, we are not concerned about the differences between these observational products in this study. Instead our motivation is the impact of SIT initialisation on seasonal predictions of Arctic sea ice cover (i.e., extent and ice-edge location). We use the – really quite comprehensive – studies of Ricker et al. here purely to provide approximate bounds below which the CS2 is likely to be of no value. However when we come to do this as a data assimilation problem – rather than a seasonal coupled prediction problem – we will be very interested in the observational uncertainty/properties.

**Changes to the manuscript:** A sentence has been added to Section 2.2.1 that states that other centres provide CS2-derived Arctic sea ice thickness products. An additional citation to Allard et al. (2018) is included (who compare the different products) as well as a citation to Stroeve et al. (2018). (P7 L7-9)

*- I don't get what this extra quality checking of the CPOM CS-2 data is. The fact you have included a personal comms from one of the data producers of that dataset makes it seem like this is something they do too? What exactly do you mean by smearing?*

This is related to the regridding and binning of the data performed by CPOM to ensure that high spatial gradients are not smeared (averaged) out. The approach taken here was that recommended by CPOM and so should be considered part of their observational processing.

**Changes to the manuscript:** This text has been removed from the revised version of the manuscript to prevent it causing a distraction to the reader.

*Why not use daily along-track CryoSat-2 data? I thought this was the whole purpose of CPOM releasing the daily along-track data? Instead they grid the data, then you grid the data, then you interpolate to a get a daily thickness?*

Yes, you are correct. Assimilating altimeter tracks of thickness (or more likely the raw freeboard) is the ultimate goal for SIT assimilation in our systems. However much work is required to do this. Observational errors need to be quantified (including representativeness), and model/observation covariances and correlations are required to spread the data from the tracks. Finally, balancing with ice concentration and other aspects of the model (SST, SSS) are required. Doing all this is a considerable undertaking and so, before doing this, we wanted to be sure that the SIT initialisation would have an impact on the model – hence this feasibility study using gridded data.

**Changes to the manuscript:** More information is now provided in the Summary section about what would be required to fully initialise GloSea sea ice thickness using the NEMOVAR 3D-Var assimilation scheme – which would use along-track altimeter data. (P17 L11+) The new hypothesis paragraph towards the end of the Introduction also includes mention that we are motivated here by wanting to assess the feasibility of SIT initialisation within GloSea (P4 L17+).

*Hard to tell what this volume comparison really means. You compare with PIOMAS but then say that data is biased low (which I'm not actually sure is true when you look at more than one CS-2 estimate) so it is actually good that you are further from that data? You say this was expected but this seems like a hindsight statement to me. I agree PIOMAS data can provide useful context but I don't agree with how you've used it. I think you should just show the CS-2 data and say look, the assimilation does what it is supposed to do.*

Yes, it was expected that the winter sea ice volume would be higher than PIOMAS. This was an obvious and logical expectation given that the CPOM CS2 data we are assimilating has higher volume than PIOMAS (as documented by the Laxon et al. and Tilling et al. CPOM studies). However, the key point here is that this doesn't actually matter. We only include PIOMAS comparison in our evaluation as a reference because it is well understood and widely used for this purpose. We do not use it for verification.

**Changes to the manuscript:** The manuscript has been changed to make it clear that we include PIOMAS as a reference only. In line with the final piece of advice here, Figure 1 has been expanded to show the impact of assimilating CS2 thickness data - 3 more panels have been added to show the modelled Oct-Apr thickness from the CTRL and ThkDA reanalyses, along with the difference between these two experiments (ThkDA-CTRL). An assessment of the impact of SIT initialisation on the sea ice volume between the CTRL and ThkDA experiments is now provided immediately after the discussion of Figure 1. To enable this, the order of Figure 2 and Figure 3 have been switched. (P9 L26+) No mention of PIOMAS is made until after this discussion and we qualify its inclusion by stating that it is used here purely as a reference because it is well understood and widely used for the purpose. (P10 L9+) Finally, the text stating that PIOMAS has been shown to underestimate thickness/volume in the winter has been changed to specifically state when compared with CPOM CS2 data. (P10 L12-15)

*P10 L10-20 and elsewhere: Very confusing to me if these are hindcasts or genuine forecasts. You use both labels interchangeably. How could you move forward to produce genuine forecasts?*

As stated earlier, these are hindcast or retrospective forecasts that are identical to genuine forecasts in everything except the fact that they are performed in the past (i.e., with the result already known). A 'hindcast' is a prediction made for a known past state whereas a 'forecast' is made now for an unknown future state. All other aspects of the prediction are essentially identical. To produce genuine forecasts we would just need initial conditions for now and then wait 4 months to see how well they did.

**Changes to the manuscript:** use of 'forecast' and 'hindcast' have been replaced with 'prediction' where appropriate throughout the manuscript. We have also added text to make it clearer that the GloSea seasonal predictions made here (and generally within the hindcast context) are done without any knowledge of future observations. (P5 L14; P11 L27-29)

*Specific comments*

*In the abstract: really the 'first time, we directly assess the impact of winter sea ice thickness initialisation on the skill of seasonal summer forecasts'? Do you mean in the Met Office model framework? I think Allard and others have done this and I also don't think you do assess forecast skill*

Yes, this is the first documented study to assess the impact of using satellite sea ice thickness data to initialise a fully coupled seasonal prediction system. This fact is confirmed by the comments of referee #2 who states, "This study represents the first known fully coupled atmosphere-ocean-ice forecast system to utilize CryoSat-2 ice thickness data for seasonal forecasts".

The study of Allard et al (2018) is very different because they use a forced ocean-sea ice model to perform long ocean/sea ice analyses. They also perform a thorough assessment of the thickness analyses produced and consider short-range, uncoupled, forecasts. Here we are only interested in the analysis in the context of providing sea ice initial conditions for our seasonal predictions – made using the GloSea seasonal prediction system.

We are a little confused about the referee's comment "I also don't think you do assess forecast skill". Here we run 24 retrospective seasonal forecasts per year, for each of 5 years (120 in total), to produce seasonal predictions of September Arctic sea ice using initial conditions in April/May. These predictions are performed under forecast conditions using an initialised coupled climate model in the same way as we do operationally in the GloSea Seasonal Prediction System. We then evaluate the quality of September-mean Arctic sea ice predictions by comparing basin-wide extent against observational estimates (from NSIDC, HadISST & OSI-SAF – the latter using the CMEMS reanalysis), and by calculating integrated ice edge error (IIEE) against the latter dataset. This is the standard methodology for examining skill in a seasonal forecast system, where the accumulation of sufficient evidence from "real time" forecasts would require substantial delays to the provision of information required to update forecast systems within a realistic period.

**Changes to the manuscript:** We have expanded the Introduction to be more explicit about our motivation for this study and to provide more details about what was done in Allard et al. (2018). In particular we now explicitly mention that previous studies have looked at analyses and short-range forecasts using externally forced ocean-sea ice models rather than seasonal forecasts using fully coupled models (as we do here). (P4 L7+) We have also added text to make it clearer that the GloSea seasonal predictions made here (and generally within the hindcast context) are done without any knowledge of future observations. (P5 L14; P11 L27-29)

*Introduction - In general you need more updated references. A lot of this discussion is a bit outdated now. i.e.:*
 *- Drop the Vaughan/IPCC refs and use the more specific refs. Try Serezze & Stroeve 2015 for a more recent seasonal sea ice trends citation?*
*- The Collins/IPCC is also a bit outdated. I think you can add some of the more recent references to sea ice projections - e.g. Jahn 2016, 2018.*

We believe the most appropriate citations are those given. The intention here is to motivate the fact that Arctic sea ice has declined/is in decline and that is projected to continue.

The most robust evidence of this is provided by these multi-author IPCC references, which are created by a multi-disciplinary (multi-centre, multi-country, etc.) team of authors. (This is also true for the very comprehensive, multi-author study of Meier et al. (2014).)

*P2 L1-11 - I think this is not useful information as it is not that relevant to seasonal forecasting and a lot of the references and discussion are outdated. Either update/improve or drop.*

We disagree and think this is extremely relevant material. Substantial resources are being invested in seasonal planning in the Arctic (e.g. within the EU's Horizon 2020 programme, the ARCUS Sea Ice for Walrus Outlook (SIWO), in support of projects endorsed by the WMO's Year Of Polar Prediction (YOPP) & MOSAiC), with more such investment by both government and private organisations likely in the future. In particular, there are a number of projects endorsed by YOPP which focus on sea ice seasonal prediction (you can see an overview of YOPP-endorsed projects at https://apps3.awi.de/YPP/endorsed/projects, noting in particular: https://apps3.awi.de/YPP/pdf/stream/79, https://apps3.awi.de/YPP/pdf/stream/100, https://apps3.awi.de/YPP/pdf/stream/106, and  https://apps3.awi.de/YPP/pdf/stream/172).

We do not believe that, as a general rule, citations should have a "best before date". So if no further information has been published in the meantime to contradict the findings of these papers, which clearly have had a large impact on the funding agencies, then they are entirely appropriate.

*P2 L13 - change sentence ordering.*

**Changes to the manuscript**: We have changed this to: "Interest in seasonal predictions has increased following the drastic reduction in Arctic sea ice extent in the summer of 2007, which led to a (then) record-low summer minimum extent being set." (P2 L13-14)

*P2 L17 - the predictive skill sentence is confusing. SIPN haven't really assessed that.*

Yes it is true that SPIN, the US project, has not done this itself. However, our point here is that the existence of SIPN has caused this as a secondary effect.

**Changes to the manuscript**: The text has been changed to make it clear that the community that has been built up around the SIO has enabled collaborative activities addressing such issues, rather than SIPN itself having done this. (P2 L18)

*P2 L19-23 - I don't think you've really said why it is interesting though! Either make a clearer point regarding its scientific interest (e.g. what the predictability/memory of the system is compared to other components of the climate system).*

**Changes to the manuscript**: we have included some text to make it clear that we are talking about the fact that seasonal forecasts are considerably longer than the (typically 1-2 week) limit, beyond which, the chaotic nature of the atmosphere and ocean inhibit traditional deterministic forecasting. We have included a new reference (Slingo and Palmer, 2011) which documents this well. (P2 L21-22)

*Also I don't think it is clear that sea ice is now necessarily harder to predict. Having some enhanced variability may be useful. Your figure 4 doesn't show an increase in ice edge error for instance!*

In fact Figure 4 does have an increasing trend! The IIEE in the control predictions (Fig 4b) is increasing over the period 1992-2015 with a small slope of approx. 0.0087 million square km per year (or 8700 square km per year). Although this is small relative to the long-term mean, it is statistically significant (p-value < 0.016). Of course, this in itself does not prove that forecasting is more challenging with less/thinner/more variable sea ice – and we are not arguing here that it does. However thinner sea ice will be more heavily influenced by the non-linear, chaotic atmospheric circulation (both dynamically and thermodynamically), which would undoubtedly be less predictable.

That prediction of sea ice becomes harder as the ice thins is one of the results of Holland et al (2011) who show that "ice area in a thicker sea ice regime generally exhibits higher potential predictability for a longer period of time". Furthermore Stroeve et al. (2014) also support this stating: "The reduced predictive skill as the winter ice cover thins has been noted in some of the contributions to the SIO and appears to be coincident with the rapid thinning of the ice cover."

**Changes to the manuscript**: We have added additional citations to Holland et al. (2011) and Stroeve et al. (2014) at the end of this sentence. (P2 L25)

*P2 L27 - why exactly does a lack of observations make the forecasts harder? Less to assimilate in models or to validate? Below you list a number of observations that are available in the poles...*

**Changes to the manuscript**: we have added "…meaning that the initial conditions used for forecasts in the Arctic are less accurate than for lower latitudes." To emphasise how a lack of observations would make forecasting harder. (P2 L31)

*P2 L35 - include acronym definitions.*

**Changes to the manuscript**: definition of acronym FOAM = "Forecast Ocean Assimilation Model has been brought forward from later in the manuscript.

*P3 L4 onwards - this is a bit of a confusing paragraph to me. What is the point you are trying to make? In general my view is that there is hope for dynamical models being used for skillful sea ice forecasting based on some of the perfect model studies that you cited. However the SIO has really shown that they are not currently performing much better than the linear trend in many cases (as shown by Stroeve et al., 2014). It still seems that dynamical models are lagging behind more simple statistical methods (e.g. Schroeder et al., 2014, Petty et al., 2017).*

*I think you need to add in some comments on the different forecast methods available, merge with the following paragraph about improving dynamical models and make clearer what the motivation of this study is! This should be the key paragraph of the introduction.*

**Changes to the manuscript:** We have replaced this paragraph to better motivate our interest in seasonal predictions using dynamical models. (P3 L7+) Additionally **t**he penultimate

paragraph of the Introduction now contains a hypothesis statement to make it clear what the motivation of the study is. In particular we hypothesise that seasonal predictions of late-summer (September) sea ice cover made using our fully coupled dynamical model, will be improved by initialising sea ice thickness in early spring (May) using observations of thick sea ice derived from CS2. (P4 L17+)

*P3 l 24 - not sure how this point links to the above.*

The motivation here is that the previous points all list examples that suggest that winter sea ice thickness would be important for seasonal forecasting of Arctic summer sea ice. This final point notes that SIT initialisation is not the complete story because model uncertainty (model structural uncertainty and model choices/parameters) is also important for the evolution of forecast errors – and likely dominate over initial condition uncertainty.

**Changes to the manuscript:** We have changed this paragraph to make the story clearer. We start by stating that several studies have shown that winter sea ice thickness provides important preconditioning for the evolution of Arctic sea ice through the summer melt season, and then say that these studies suggest that SIT could improve seasonal forecasts of summer sea ice. We then note that, on seasonal timescales, model uncertainty is likely to dominate the evolution of errors through the forecast. (P3 L19-33)

*P4 L1-4 - but in the abstract you imply you are the first to do this?! I guess you meant in your fully coupled Met Office forecast framework. You need to make that clearer.*

Yes we are. As stated previously this is the first use of satellite thickness data to initialise seasonal coupled predictions of Arctic sea ice. The previous studies listed here are using forced ocean-sea ice models and performing reanalyses and/or short-range forecasts. In contrast we are performing seasonal predictions (5-month forecasts) using a fully coupled atmosphere-ocean-sea ice-land model with the GloSea seasonal prediction system.

**Changes to the manuscript:** We have expanded the Introduction to be more explicit about our motivation for this study and to provide more details about what was done in Allard et al. (2018). In particular we now explicitly mention that previous studies have looked at analyses and short-range forecasts using externally forced ocean-sea ice models rather than seasonal forecasts using fully coupled models (as we do here). (P4 L17+)

*P5 L11-13 - why just mention the ocean reanalysis component here? Would be more understandable if you referred to GloSea as a reanalysis.*

No! GloSea is a coupled seasonal prediction system (see MacLachlan et al (2014)). Referring to it as a reanalysis would be more misleading.

The motivation in this section of the manuscript is to introduce the fact that long reanalyses are performed using an offline analogue of the FOAM ocean analysis system. These long reanalyses are primarily used within the GloSea seasonal prediction system to initialise hindcast (or retrospective forecast) experiments. However they have also been utilised by several studies to help further understanding of the ocean and how the ocean has changed over the satellite period (e.g. within ORA-IP). As the ocean reanalysis is used within the GloSea seasonal prediction system, it is sometimes (erroneously/unfortunately) referred to as

the GloSea ocean re-analysis. However, GloSea is a coupled global seasonal prediction system (see MacLachlan et al., 2014).

**Changes to the manuscript**: We have changed several instances of "GloSea" to "GloSea seasonal prediction system" to help make it clearer that GloSea is much larger than an ocean reanalysis.

*P5 L23 - I don't think these are the correct citations here. Link to relevant passive microwave concentration datasets instead or recent papers describing that long-term record (e.g. Parkinson/Comiso papers).*

The motivation behind these citations is two-fold: 1) to provide references for the HadISST and NSIDC data sources that we use within our forecast evaluation; 2) to show that we have long-term sea ice concentration observations from multiple sources. The citations given here are those recommended/requested by the data providers for the NSIDC sea ice index (https://nsidc.org/data/g02135) and for the HadISST dataset (https://www.metoffice.gov.uk/hadobs/hadisst/). Therefore, their inclusion is required for our objective #1 and we re-use them for our objective #2 to avoid including lots of similar citations.

*P5 L25 - should reference Kwok and Cunningham 2008 instead.*

We shall change this from the 2009 reference to the Kwok and Cunningham (2008) reference.

**Changes to the manuscript**: Kwok and Cunningham (2008) manuscript has been added to the references and is cited here in place of Kwok et al. (2009). (P6 L18)

*P6 L16-17 - what do you mean by sensitive here? I think you mean uncertain/challenging? Again, is there nothing in one of the CPOM papers that highlights this issue?*

When freeboard is very low it is difficult to distinguish from SSH fluctuations and gravity waves. This issue is well documented in the comprehensive Ricker et al. citations that we have used. Given that the issue is related to the whole process of deriving freeboard from satellite altimetry – in particular CS2 SIRAL radar altimeter which penetrates the snow and into the upper layer of the ice – rather than the centre who happen to be processing the data, we are happy that these citations cover the issue adequately.

**Changes to the manuscript**: The word 'sensitive' has been replaced with 'uncertain'. (P7 L15)

*P6 L30 - I think this is a big guess. Do we really know much ice < 2 m melts away each summer? That Keen modeling study (Fig 2?) suggests some 40% of the ice less than 2 m (including ice and snow in that thickness) does not melt through in summer. A lot will have to do with how much snow there was on the ice and where/when melt onset occurs. Even if it does melt away, this seems to be crucial information for determining solar absorption that can drive SST increases and further sea ice melt. Also need to make the point here (and earlier) that AWI do produce a merged product!*

**Changes to the manuscript**: We do not undertake a summary, or comparison, of available sea ice thickness datasets in this study. To avoid this sort of confusion, the abovementioned

paragraph has been removed in the revised manuscript. We now only refer to CS2 thickness observations in Section 2.2.1 ("CryoSat-2 thickness observations"). The new hypothesis/motivation paragraph in the Introduction, will tell the reader that our interest is initialising thick sea ice using CS2 and testing the impact on seasonal predictions of September sea ice cover.

*Section 2.3 - Why do you need extent and concentration? Surely you are just assimilating sea ice concentration? This needs to be made much clearer here.*

*There appears to be some confusion that this section is referring only to data used for assimilation. Section 2 is about all the models and datasets used in this study – both for assimilation and for evaluation. Section 2.3 covers the validation datasets in paragraph 1 and the assimilation data in paragraph 2.*

**Changes to the manuscript:** Section 2.3 has been modified to include an introductory sentence that explains that data is used for assimilation and evaluation purposes. (P7 L27-28) New subsection headings (2.3.1 and 2.3.2) are also added to split the introduction of observations according to whether they are used for evaluation (2.3.1), or assimilation (2.3.2). (P7 L29; P8 L12)

*- The use of CMEMS 'data' seems very confusing to me. Why not use observed ice concentration?!*

It is very common to use an analysis product for evaluation purposes. For example, many people use the OSTIA analysis for evaluating SST or ERA-Interim for evaluating atmospheric variables. Likewise, people are now starting to use the blended CS2SMOS product (Ricker et al. 2017) to evaluate sea ice thickness (which itself is an SIT analysis created using O/I assimilation methods). Although we would never assimilate analysis products like these, we do use them for the purpose of model evaluation.

In this study we validate using the ice concentration from the CMEMS reanalysis, which is an analysis made using the OSI-SAF sea ice concentration data. The main reason for using this is that it is already on the correct model grid. We state that: "Using this CMEMS reanalysis has the benefit that it is performed on the same ORCA025 grid as the ocean-sea ice components of the GloSea seasonal forecasting system, which makes spatial comparisons easier."

It is important to note that, except for the differences due to the spatial resolution, the sea ice concentrations in the OSI-SAF observed data and the CMEMS analysis are virtually identical. The CMEMS product can therefore be thought of as a dynamically consistent re-gridding. It is of course very important to compare extent using products on the same grid because extent, as a metric, is very much dependent on grid/resolution (Notz, 2014).

*- The NSIDC sea ice index is just a monthly index of total ice extent. This isn't what you use, right?*

Yes we use the NSIDC single number extent for the purposes of validating our September-mean ice extent predictions (in Fig 4). However we also use extent derived from HadISST sea ice concentration, and from the CMEMS reanalysis (i,e., OSI-SAF).

*P8 L6 - what is the size of your model grid (in kilometers?) how does this translate to the 5 km CPOM data?*

The ORCA025 tripolar grid was created to avoid the singularity associated with the convergence of meridians at the North Pole, which it achieves by defining two distinct north poles over Canada & Siberia. The ocean points in between form a variable resolution grid with highest resolution nearest the two poles i.e., in the Canadian Arctic Archipelago and the Laptev Sea. The resolution in the Arctic Ocean ranges from ~9km up to ~15km.

**Changes to the manuscript:** Section 2.1 has been expanded to include more details of the sea ice component of the model (as requested by Referee #2). We have included details of the grid resolution within this text as follows: "ranging from 8.9 km to 15.5 km in the Arctic Ocean basin" (P6 L5-6)

*P8 L14-24 - Think you should list out the CICE thickness categories. How else could you have done this? I think it would be worth presenting more sophisticate approaches for future work, however I get that you started with this simple approach.*

We use standard WMO categories that are one of default options within the CICE model. These are listed in the supporting Blockley et al., (2014), and Ridley et al. (2018) references.

**Changes to the manuscript:** Section 2.1 has been expanded to include more details of the sea ice component of the model (as requested by Referee #2). We have included details of the model thickness categories within this text as follows: "(lower bounds: 0, 0.6, 1.4, 2.4 and 3.6 m)". (P6 L10) More details are provided as future plans for sea ice thickness assimilation in the Summary section (P17 L11+)

*Figure 2 - I think you should also show (maybe in the supplementary info?) what the pre nudged, and nudged thickness fields are. Could just do this for the mean October-April thickness and also update Figure 1 to show this longer season too. If the mean thickness was way off before it makes sense that assimilating the thickness will improve things.*

**Changes to the manuscript**: Figure 1 has been expanded to show the impact of assimilating CS2 thickness data on the sea ice initial conditions. In addition to the original plot showing the CS2 thickness data, 3 more panels have been added to show the modelled thickness from the control (CTRL) and thickness initialised (ThkDA) reanalyses, along with the difference between these two experiments (ThkDA-CTRL). In line with the referee's suggestion, the time-period has been increased from DJF to October-April mean. A new paragraph has been added at the start of Section 3.1 to describe the general impact of CS2 initialisation on the winter thickness. (P9 L26+)

*Why the different start dates for the forecasts? Pretty confusing.*

This is how the GloSea seasonal prediction system works. GloSea runs every day at the Met Office and produces 2 forecasts of length 210-days. These are used with forecasts from previous days to create a large, lagged ensemble of forecasts. For the hindcasts/reforecasts performed here, an 8-member ensemble is run for fewer distinct start dates. This is done to

be consistent with the way that GloSea performs its operational hindcasts – which are initialised from 4 start dates per month (see MacLachlan et al).

*Figure 4 is hard to see. Maybe box plots of the recent years showing the variability in the different estimates in the different years?*

Referee #2 also asked for the layout of Figure 4 to be modified and so it has been overhauled.

**Changes to the manuscript**: Figure 4 has been changed to make it easier to read. More space on the plot is now devoted to the key time-period of 2011-2015, and spacing has been used to ensure that the individual ensemble member values are now distinguishable from each other. The figure caption has been modified to make it clear that these changes to the spacing are done purely for ease of viewing. (P32)

*P11 L10 - CMEMS isn't really an observational estimate, right? Based on the assimilation of OSI-SAF...*

**Changes to the manuscript**: this text has been modified to make it clear that we compare with NSIDC and HadISST observational datasets as well as the CMEMS reanalysis dataset. (P12 L22-24)

*Pll L14 - not sure what you mean by 'building a picture' here. I see no value in showing that earlier data.*

**Changes to the manuscript**: Text has been added to better paint a picture of the errors in the control simulations. Discussion of the extent predictions has been expanded to details how the control experiment predictions of extent are consistently biased low. (P12 L28-30) An additional sentence has been added to the IIEE discussions to describe that the ice edge error is consistent through the 24 year period covered by the control simulations. (P13 L15-17)

*P11 L17 what is close*

**Changes to the manuscript**: "close" has been quantified as "within 0.12 $\times 10^6$ km$^2$". (P13 L1)

*Figure 5 - include numbers on map. Plot the IEEE as a time series.*

**Changes to the manuscript**: Figure 5 has been modified so that each panel includes a box containing the observed and modelled extent along with the IIEE. The figure caption has been modified accordingly. (P33-34)

*Figure 7 and 8 - I don't understand these maps. How exactly is the data shown in Figure 8 calculated? Also is Figure 8 averaged over the entire year, but figure 7 is September? Why are the pressure units different? Perhaps better to show Figure 7a and 8a together, then 8a and 8b*
*The discussion of using ensemble members instead of the ensemble means and how this relates to assessing model bias was confusing and needs more description/clarification.*

**Changes to the manuscript**: Figures 7 and 8 have been merged together (new Figure 8 - P36) to include both differences in the mean fields (left) and the forecast errors (right). We now also show z500 and MSLP for both. Section 4.2 has been modified to provide better motivation for Figure 8. In particular, and as requested, Section 4.2 paragraph 2 provides a thorough overview of what is plotted in Figure 8. (P14 L22+)

(NB. in the original manuscript "September" was accidentally left off the Figure 8 caption, which prompted the question about whether this plot was the entire year. This has been fixed as part of the changes described here.)

*Should either enhance the CLIM analysis or drop. i.e. repeat with using a fixed (think FIXED_IC would be a better acronym) for all years of data available.*

**Changes to the manuscript**: "CLIM-2015" is changed to "FIXED-IC" throughout the manuscript.

*How do these results compare with Allard?*

Our study is very different from Allard et al. As stated previously, we are initialising a fully-coupled atmosphere-ocean-sea ice-land (AOIL) model with CryoSat-2 (CS2) sea ice thickness and using it to perform an ensemble of seasonal predictions from May through to September. We find that initialising with CS2 SIT gives us a considerable improvement in our seasonal predictions of Arctic sea ice extent and ice edge location. We also show that memory of winter thickness changes in the initialisation carry through to the end of summer.

Meanwhile Allard et al. (2018) performed an 18-month reanalysis using an externally forced ocean-sea ice model forced by atmospheric analyses. They assimilated CS2 thickness (from multiple sources) during the winter when the data is available. In contrast to our study, and by virtue of the fact they run a reanalysis, they continue assimilating all other variables throughout the year. They show a considerable improvement in their winter thickness analyses when using CS2.

What the two studies have in common is that:

1) we both show that assimilating sea ice thickness provides an improvement (us: in coupled seasonal forecasts; them: in their externally forced reanalysis);
2) we both show that memory of initialised winter thickness is still present in the summer (us: after a 5-month free-running coupled model forecast; them: after 5 months of assimilating everything except SIT in an externally forced reanalysis).

Both studies show that sea ice thickness initialisation is beneficial – albeit in very different setups.

**Changes to the manuscript:** We have expanded the Introduction to be more explicit about our motivation for this study and to provide more details about what was done in Allard et al. (2018). In particular we now explicitly mention that previous studies have looked at analyses and short-range forecasts using externally forced ocean-sea ice models rather than seasonal forecasts using fully coupled models (as we do here). (P4 L17+)

*P14 L21 - not sure what you mean by work well. Maybe not gone wrong?*

Technically, the application of thickness increments within the CICE sea ice model while performing a sea ice analysis has worked exactly as expected. The increments have been retained by the model and produce thickness initial conditions, for initialising the seasonal forecasts, that are much closer to the CS2 data. There is also the matter of the exceptional improvement to sea ice location (IIEE) when initialising with SIT.

**Changes to the manuscript**: An extra sentence has been added here to quantify this further. We now state that the winter thickness initial conditions, generated using the sea ice thickness nudging, are much closer to the CS2 thickness observations as illustrated by the new panels in Figure 1. We also state that these thickness changes lead to considerable improvement in skill when used to initialise GloSea seasonal predictions. (P17 L1-5) The existing text, which stated that (importantly) the model is able to retain the thickness information, has also been retained.

*P14 L1 Why is this from dynamics not thermodynamics?*

The distribution of thickness across the Arctic is caused by a series of dynamic processes. The ice motion, primarily driven by the winds, consists of a re-circulation around the Beaufort Gyre with a transpolar drift which drives across from Siberia to the north coast of Greenland and north of Svalbard. This causes thick ice to pile up north of Greenland. The fact that we have thicker ice dragged around into the Beaufort Sea and not enough thin ice north of Fram Strait suggests that the ice is too mobile. Both the proposed EAP rheology and form-drag changes would reduce the ice speed and should reduce the bias.

There is of course a small chance that this is thermodynamically driven, which is why we say this is "most likely" caused by deficiencies in the dynamics.

*Also drop 'so-called ice-ice force'*

**Changes to the manuscript**: this text has been removed.
* * *
**Response to review #2**

**Specific Comments**

*Page 3 Line 1: Add Cummings and Smedstad (2014) for another coupled ocean-ice modeling system with data assimilation, here is a full reference: Cummings, J. A. and O. M. Smedstad, 2014: Ocean Data Impacts in Global HYCOM, Journal of Atmospheric and Oceanic Technology, 31, doi:10.1175/JTECH-D-14-00011.1*

The articles cited here (i.e., Tonani et al., Martin et al., Balmaseda et al.) are all large multi-centre papers which describe all of the global ocean/ice analysis + forecasting systems that contribute to the GODAE OceanView project. This includes HYCOM-based systems such as that documented in Cummings and Smedstad (2014). Jim Cummings is part of the author list for one of these, and several Cummings papers are cited. However, this exact paper is not one of those referenced in the papers we cite here.

**Changes to the manuscript**: Cummings and Smedstad (2014) citation added.

*Page 7 line 31: why couldn't a longer period, say from 2010-2017 or at least 2010-2016 be used?*

This work was started in 2016 and so the analysis was performed up to September 2015, which was the most recent summer period at that time. This is true for all the control runs that we used in this project as well as the CS2 runs that we performed ourselves. Whilst undertaking this work, we were only able to perform such a large ensemble of seasonal predictions because the delay finalising our model configuration for CMIP6 meant that there was a good chunk of computational resource available. This is no longer the case and CMIP6 runs are in full swing now – meaning that we do not have the computational resources available to extend this to 2016 or 2017 at this time.

*Page 8 line 14: which version of CICE (v4.0, 4.1) is used. Are melt ponds used in this study?*

CICE vn5.1.2 is used here and the configuration includes topographic melt ponds. The coupled model version used here (GC3) is documented in Williams et al. (2017) and the sea ice component therein (GSI8) is documented in more detail in Ridley et al. (2018). Although we do not intend to include lots of technical information about the sea ice model used, we do agree with the referee that more information is required here.

**Changes to the manuscript:** Section 2.1 has been expanded to include more details of the sea ice component of the model. This text includes details about CICE model version and key features of the sea ice model component such as multi-layer thermodynamics, 5 thickness categories, and prognostic melt-ponds. We also included details of the grid resolution and category bounds as requested by Referee #1. (P6 L5+)

*Page 14 line 17-18: can you quantify bias reduction with some range or percentages?*

We have done this in the revised manuscript. However, we feel that this sits better at the end of page 9 where the bias/difference is first discussed rather than on page 14.

**Changes to the manuscript:** In the revised manuscript, the change in long-term mean thickness has been quantified by specifying percentage changes and explicit values. This is done for the whole Arctic as well as separately for the Atlantic and Pacific sectors (P11 L2-6) Additionally, Figure 1 has been expanded to show the impact of assimilating CS2 thickness data on the sea ice initial conditions. In addition to the original plot showing the CS2 thickness data, 3 more panels have been added to show the modelled thickness from the control (CTRL) and thickness initialised (ThkDA) reanalyses, along with the difference between these two experiments (ThkDA-CTRL). In line with the referee's suggestion, the time-period has been increased from DJF to October-April mean. A new paragraph has been added at the start of Section 3.1 to describe the general impact of CS2 initialisation on the winter thickness. (P9 L26+)

*Figure 4: Please add an inset for both plots showing a blow-up for the period 2010-2015? It's a bit difficult to see with the longer data record shown.*

Referee #1 also asked for the layout of Figure 4 to be modified and so it has been overhauled.

**Changes to the manuscript**: Figure 4 has been changed to make it easier to read. More space on the plot is now devoted to the key time-period of 2011-2015, and spacing has been used to ensure that the individual ensemble member values are now distinguishable from each other. The figure caption has been modified to make it clear that these changes to the spacing are done purely for ease of viewing. (P32)

*There is no mention of ice drift in the paper. Could you analyze IABP ice drift data (Pan-Arctic domain) to determine the impact of assimilating CS2 data into the seasonal forecasts? This would complement your existing study. SIDFex is presently examining modeling center's skill in making long-term ice drift trajectory forecasts.*

We have compared the May and September sea ice velocity fields from our two main experiments (CTRL and thkDA). We find that the May velocities are virtually indistinguishable in each of the 5 years (2011-2015). This is consistent with the findings of Allard et al. (2018) who show little impact on ice drift in their reanalysis comparisons. The September velocity fields are also very similar. Although slight differences arise from the differing ice coverage, the ice drift is broadly/qualitatively unchanged in the experiment using sea ice thickness initialisation.

**Changes to the manuscript**: We have included a small paragraph to state that the drift is unchanged (not shown) and that this is consistent with the findings of Allard et al. (2018). (P11 L9-11)

*I would like to see the ice edge error metric used to examine the regional differences seen from use of the CS2 data. Can it be divided into the following (or similar) basins (Beaufort/Chukchi/Bering Sea, Canadian Archipelago, Greenland Sea, Laptev Sea, Barents Sea, East Siberian Sea)?*

**Changes to the manuscript**: A new figure (Figure 6) has been added which includes details of the observational extent (from the CMEMS reanalysis), the model predictions of extent from both the CTRL and ThkDA experiments, and the ide edge error (IIEE). This is done for 3 domains which bisect the Arctic Ocean: the central Arctic, the Siberian shelf (combined Laptev, Kara + East Siberian Seas), and the combined Beaufort + Chukchi Seas. Also reported are the same statistics for the Atlantic and Pacific sectors of the Arctic. A few sentences of text has been added to Section 4.1 to discuss these results - in particular that improvement is most notable in the Atlantic sector. (P13 L31+)

*No comparisons are made against ice thickness observations from either ice mass balance buoys and/or moored ULS data. I recommend inclusion of some time series plots of the modeled ice thickness beginning with the Apr/May initializations through September for 2010-2015, with the control run included. The ensemble spread can be shown as well. This should clearly show the impact of the inclusion of the CS2 data.*

While comparison with ULS would be interesting, we feel it is out of scope for this study where we are heavily focussed on improvement to sea ice cover (extent and edge location). However, this sort of comparison is something we would wish to do before implementing a proper 3D-Var sea ice thickness assimilation scheme – and in fact is currently being undertaken within the H2020 SEDNA project.

*Lots of acronyms are used without spelling them out. A partial list is shown below. Perhaps a list or table of acronyms would be useful.*

We apologise for this oversight. The revised manuscript will ensure that acronyms are spelled out at the point of first use. We do not think that this manuscript is "acronym heavy" enough to need a glossary appendix/table though. However, we can include one if the referee (or the journal typesetters) feels strongly about this.

**Changes to the manuscript**: many changes have been made to define acronyms at 1st point of use.

**Technical Corrections**

*Page 2 line 6: replace "Better knowledge" with "Improved knowledge"*

*Page 2 line 34: define SLA here*

*Page 3 line 22: define CFSv2*

*Page 3 line 25: replace "find" with "found"*

*Page 3 Line 34: Yang et al. reference not listed in References*

*Page 4 line 3: spell out NRL (Naval Research Laboratory)*

*Page 5 line 3: spell out FGAT*

*Page 5 lines 12-14: "used" appears in sentence 3 times. Perhaps change second mention of this word to "utilized"*

*Page 5 line 26: reword statement "have been around for some years"*

*Page 5 line 27: replace "main" with "primary"*

*Page 5 line 32: spell out SIRAL*

*Page 6 line 22: replace "was" to "is"*

*Page 7 line 16: spell out SSMI/S*

*Page 7 line 28: Ridley reference says (2017, in review); reference section states 2018*

*Page 9 line 24: reword to "although IT has".*

*Page 10 line 4: delete "the fact"*

*Page 10 line 17: Williams reference in references section says 2018*

*Page 10 line 25: MacLachlan reference says 2014 in references section*

*Page 12 line 2: delete "down"*

*Page 13 line 26: reword "doing things this way"*

*Page 14 line 28: spell out SEDNA*

*Page 15 line 4: Neither Tsamodos reference is listed in References section*

*Page 15 line 17: replace "down" with "due"*

*Page 21: Peterson reference should be 2015 not 2014*

Many thanks for providing such a thorough list of technical changes.

**Changes to the manuscript**: all of these minor corrections have been made in the revised manuscript.

*Page 10 line 7: reword "amongst other things" what things?*

Essentially the "other things" mentioned here are all of the model prognostics and diagnostics which are far too many to list.

**Changes to the manuscript**: We have removed mention of "amongst other things" to prevent it causing confusion.

[revised manuscript text omitted]

**Figure 6: Mean September Arctic sea ice extent for 2011-2015 from the CMEMS reanalysis (using OSI-SAF) compared with modelled extent and ice edge error (IIEE) from the CTRL and ThkDA seasonal predictions (units x10$^6$ km$^2$). Data are shown for 3 regions distinguished by the underlying shading and corresponding box colours: combined Beaufort + Chukchi Seas (yellow), combined Kara, Laptev and East Siberian Seas (dark blue) and the central Arctic (red). Also shown (pink boxes) are corresponding statistics for the Atlantic and Pacific sectors of the Arctic Ocean, defined by splitting the Arctic Ocean (i.e., red + yellow + dark blue) along 30°W and 140°E longitude (yellow lines) – which roughly follows the Lomonosov Ridge.**

**Figure 7: Thickness difference (m) between ThkDA and CTRL experiments in May and September 2012 (contour shading) with differences calculated as ThkDA – CTRL in each case. (left) shows the difference between the reanalyses fields used to initialise the seasonal predictions on 1st May 2012. (right) shows ensemble mean forecast differences for September 2012. (Note the different scales used for the coloured shading). Overlain on both panels are September-mean contours of 15% ice concentration to represent the sea ice edge for the CTRL-HC (grey) and ThkDA-HC (pink) experiments along with the CMEMS reanalysis product (black).**

[Figure]

[Figure]

**Figure 8: Differences in mean fields (left) and root-mean-square-error (RMSE; right) between the ThkDA and CTRL** September predictions averaged over all ensemble members for the 5-year period 2011-2015. **Fields shown are: (top) near-surface 2m air temperature (T2M; K); (middle) mean sea-level-pressure (MSLP: hPa); (bottom) 500 hPa geopotential height (z500; m).** Differences are calculated as ThkDA – CTRL meaning that areas of blue (red) denote that ThkDA predictions/RMSEs are lower (higher). Black contours and hatching denote areas where differences are significant at the 95% level as determined using a Mann-Whitney U-test. **Further details can be found at the beginning of Section 4.2.**